# Nested Graph Neural Networks

**Muhan Zhang**[1,2,*]  **Pan Li**[3,†]
[1]Institute for Artificial Intelligence, Peking University
[2]Beijing Institute for General Artificial Intelligence
[3]Department of Computer Science, Purdue University

## Abstract

Graph neural network (GNN)'s success in graph classification is closely related to the Weisfeiler-Lehman (1-WL) algorithm. By iteratively aggregating neighboring node features to a center node, both 1-WL and GNN obtain a node representation that encodes a *rooted subtree* around the center node. These rooted subtree representations are then pooled into a single representation to represent the whole graph. However, rooted subtrees are of limited expressiveness to represent a nontree graph. To address it, we propose Nested Graph Neural Networks (NGNNs). NGNN represents a graph with *rooted subgraphs* instead of rooted subtrees, so that two graphs sharing many identical subgraphs (rather than subtrees) tend to have similar representations. The key is to make each node representation encode a subgraph around it more than a subtree. To achieve this, NGNN extracts a local subgraph around each node and applies a *base GNN* to each subgraph to learn a subgraph representation. The whole-graph representation is then obtained by pooling these subgraph representations. We provide a rigorous theoretical analysis showing that NGNN is strictly more powerful than 1-WL. In particular, we proved that NGNN can discriminate almost all $r$-regular graphs, where 1-WL always fails. Moreover, unlike other more powerful GNNs, NGNN only introduces a constant-factor higher time complexity than standard GNNs. NGNN is a plug-and-play framework that can be combined with various base GNNs. We test NGNN with different base GNNs on several benchmark datasets. NGNN uniformly improves their performance and shows highly competitive performance on all datasets.

## 1 Introduction

Graph is an important tool to model relational data in the real world. Representation learning over graphs has become a popular topic of machine learning in recent years. While network embedding methods, such as DeepWalk [1], can learn node representations well, they fail to generalize to whole-graph representations, which are crucial for applications such as graph classification, molecule modeling, and drug discovery. On the contrary, although traditional graph kernels [2–7] can be used for graph classification, they define graph similarity often in a heuristic way, which is not parameterized and lacks some flexibility to deal with features.

In this context, graph neural networks (GNNs) have regained people's attention and become the state-of-the-art graph representation learning tool [8–17]. GNNs use message passing to propagate features between connected nodes. By iteratively aggregating neighboring node features to the center node, GNNs learn node representations encoding their local structure and feature information. These node representations can be further pooled into a graph representation, enabling graph-level tasks such as graph classification. In this paper, we will use *message passing GNNs* to denote this class

---

*Corresponding author: Muhan Zhang (`muhan@pku.edu.cn`).
†Pan Li contributes Sec. 3.3 that proves the Theorem 1 and some implementation ideas.

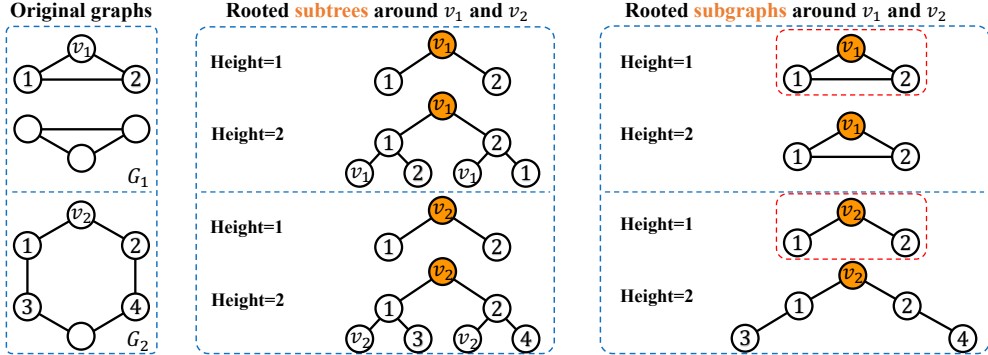

Figure 1: The two original graphs $G_1$ and $G_2$ are non-isomorphic. $G_1$ is composed of two triangles, while $G_2$ is a hexagon. However, both 1-WL and message passing GNNs cannot differentiate them, since all nodes in the two graphs share identical rooted subtrees at any height (see the rooted subtrees around $v_1$ and $v_2$ in the middle block for example). In comparison, we can discriminate the two graphs by comparing their height-1 rooted subgraphs around any nodes. For example, the height-1 rooted subgraph around $v_1$ is a closed triangle, but the height-1 rooted subgraph around $v_2$ is an open triangle (see the red boxes in the right block).

of GNNs based on repeated neighbor aggregation [18], in order to distinguish them from some high-order GNN variants [19–21] where the effective message passing happens between high-order node tuples instead of nodes.

GNNs' message passing scheme mimics the 1-dimensional Weisfeiler-Lehman (1-WL) algorithm [22], which iteratively refines a node's color according to its current color and the multiset of its neighbors' colors. This procedure essentially encodes a rooted subtree around each node into its final color, where the rooted subtree is constructed by recursively expanding the neighbors of the root node. One critical reason for GNN's success in graph classification is because two graphs sharing many identical or similar rooted subtrees are more likely classified into the same class, which actually aligns with the inductive bias that two graphs are similar if they have many common substructures [23].

Despite this, rooted subtrees are still limited in terms of expressing **all possible substructures** that can appear in a graph. It is likely that two graphs, despite sharing a lot of identical rooted subtrees, are not similar at all because their other substructure patterns are not similar. Take the two graphs $G_1$ and $G_2$ in Figure 1 as an example. If we apply 1-WL or a message passing GNN to them, the two graphs will always have the same representation no matter how many iterations/layers we use. This is because **all** nodes in the two graphs have identical rooted subtrees across **all** tree heights. However, the two graphs are quite different from a holistic perspective. $G_1$ is composed of two triangles, while $G_2$ is a hexagon. The intrinsic reason for such a failure is that rooted subtrees have limited expressiveness for representing general graphs, especially those with cycles.

To address this issue, we propose Nested Graph Neural Networks (NGNNs). The core idea is, instead of encoding a rooted subtree, we want the final representation of a node to encode a **rooted subgraph** (local $h$-hop subgraph) around it. The subgraph is not restricted to be of any particular graph type such as tree, but serves as a general description of the local neighborhood around a node. Rooted subgraphs offer much better representation power than rooted subtrees, e.g., we can easily discriminate the two graphs in Figure 1 by only comparing their height-1 rooted subgraphs.

To represent a graph with rooted subgraphs, NGNN uses **two** levels of GNNs: base (inner) GNNs and an outer GNN. By extracting a local rooted subgraph around each node, NGNN first applies a base GNN to each node's subgraph independently. Then, a subgraph pooling layer is applied to each subgraph to aggregate the intermediate node representations into a subgraph representation. This subgraph representation is used as the final representation of the root node. Rather than encoding a rooted subtree, this final node representation encodes the local subgraph around it, which contains more information than a subtree. Finally, all the final node representations are further fed into an outer GNN to learn a representation for the entire graph. Figure 2 shows one NGNN implementation using message passing GNNs as the base GNNs and a simple graph pooling layer as the outer GNN.

One may wonder that the base GNN seems to still learn only rooted subtrees if it is message-passing-based. Then why is NGNN more powerful than GNN? One key reason lies in the subgraph pooling

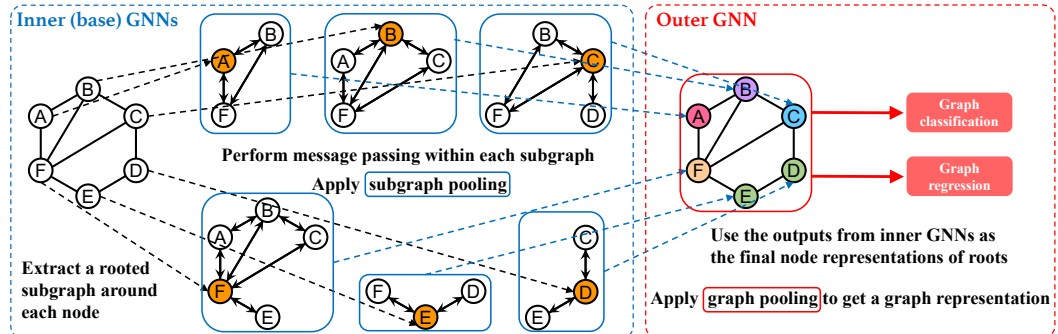

Figure 2: A particular implementation of the NGNN framework. It first extracts (copies) a rooted subgraph (height=1) around each node from the original graph, and then applies a base GNN with a subgraph pooling layer to each rooted subgraph independently to learn a subgraph representation. The subgraph representation is used as the root node's final representation in the original graph. Then, a graph pooling layer is used to summarize the final node representations into a graph representation.

layer. Take the height-1 rooted subgraphs (marked with red boxes) around $v_1$ and $v_2$ in Figure 1 as an example. Although $v_1$ and $v_2$'s height-1 rooted subtrees are still the same, their neighbors (labeled by 1 and 2) have different height-1 rooted subtrees. Thus, applying a one-layer message passing GNN plus a subgraph pooling as the base GNN is sufficient to discriminate $G_1$ and $G_2$.

The NGNN framework has multiple exclusive advantages. Firstly, it allows freely choosing the base GNN, and can enhance the base GNN's representation power in a plug-and-play fashion. Theoretically, we proved that NGNN is more powerful than message passing GNNs and 1-WL by being able to discriminate almost all $r$-regular graphs (where 1-WL always fails). Secondly, by extracting rooted subgraphs, NGNN allows augmenting the initial features of a node with subgraph-specific structural features such as distance encoding [24] to improve the quality of the learned node representations. Thirdly, unlike other more powerful graph neural networks, especially those based on higher-order WL tests [19–21, 25], NGNN still has linear time and space complexity w.r.t. graph size like standard message passing GNNs, thus maintaining good scalability. We demonstrate the effectiveness of the NGNN framework in various synthetic/real-world graph classification/regression datasets. On synthetic datasets, NGNN demonstrates higher-than-1-WL expressive power, matching very well with our theorem. On real-world datasets, NGNN consistently enhances a wide range of base GNNs' performance, achieving highly competitive results on all datasets.

## 2 Preliminaries

### 2.1 Notation and problem definition

We consider the graph classification/regression problem. Given a graph $G = (V, E)$ where $V = \{1, 2, \ldots n\}$ is the node set and $E \subseteq V \times V$ is the edge set, we aim to learn a function mapping $G$ to its class or target value $y$. The nodes and edges in $G$ can have feature vectors associated with them, denoted by $\boldsymbol{x}_i$ (for node $i$) and $\boldsymbol{e}_{ij}$ (for edge $(i, j)$), respectively.

### 2.2 Weisfeiler-Lehman test

The Wesfeiler-Lehman (1-WL) test [22] is a popular algorithm for graph isomorphism checking. The classical 1-WL works as follows. At first, all nodes receive a color 1. Each node collects its neighbors' colors into a multiset. Then, 1-WL will update each node's color so that two nodes get the same new color if and only if their current colors are the same and they have identical multisets of neighbor colors. Repeat this process until the number of colors does not increase between two iterations. Then, 1-WL will return that two graphs are non-isomorphic if their node colors are different at some iteration, or fail to determine whether they are non-isomorphic. See [7, 26] for more details.

1-WL essentially encodes the rooted subtrees around each node at different heights into its color representations. Figure 1 middle shows the rooted subtrees around $v_1$ and $v_2$. Two nodes will have the same color at iteration $h$ if and only if their height-$h$ rooted subtrees are the same.

# 3 Nested Graph Neural Network

In this section, we introduce our Nested Graph Neural Network (NGNN) framework and theoretically demonstrate its higher representation power than message passing GNNs.

## 3.1 Limitations of the message passing GNNs

Most existing GNNs follow the message passing framework [18]: given a graph $G$, each node's hidden state $\boldsymbol{h}_v^{t+1}$ is updated based on its previous state $\boldsymbol{h}_v^t$ and the messages $\boldsymbol{m}_v^{t+1}$ from its neighbors

$$\boldsymbol{h}_v^{t+1} = U_t(\boldsymbol{h}_v^t, \boldsymbol{m}_v^{t+1}), \quad \text{where} \quad \boldsymbol{m}_v^{t+1} = \sum_{u \in N(v|G)} M_t(\boldsymbol{h}_v^t, \boldsymbol{h}_u^t, \boldsymbol{e}_{vu}). \tag{1}$$

Here $M_t, U_t$ are the message and update functions at time stamp $t$, $\boldsymbol{e}_{vu}$ is the feature of edge $(v, u)$, and $N(v|G)$ is the set of $v$'s neighbors in graph $G$. The initial hidden states $\boldsymbol{h}_v^0$ are given by the raw node features $\boldsymbol{x}_v$. After $T$ time stamps (iterations), the final node representations $\boldsymbol{h}_v^T$ are summarized into a whole-graph representation with a readout (pooling) function $R$ (e.g., mean or sum):

$$\boldsymbol{h}_G = R(\{\boldsymbol{h}_v^T | v \in G\}). \tag{2}$$

Such a message passing (or neighbor aggregation) scheme iteratively aggregates neighbor information into a center node's hidden state, making it encode a local rooted subtree around the node. The final node representations will contain both the local structure and feature information around nodes, enabling node-level tasks such as node classification. After a pooling layer, these node representations can be further summarized into a graph representation, enabling graph-level tasks. When there is no edge feature and the node features are from a countable space, it is shown that message passing GNNs are at most as powerful as the 1-WL test for discriminating non-isomorphic graphs [27, 19].

For an $h$-layer message passing GNN, it will give two nodes the same final representation if they have identical **height-$h$ rooted subtrees** (i.e., both the structures and the features on the corresponding nodes/edges are the same). If two graphs have a lot of identical (or similar) rooted subtrees, they will also have similar graph representations after pooling. This insight is crucial for the success of modern GNNs in graph classification, because it aligns with the inductive bias that two graphs are similar if they have many common substructures. Such insight has also been used in designing the WL subtree kernel [7], a state-of-the-art graph classification method before GNNs.

However, message passing GNNs have several limitations. Firstly, rooted subtree is only one specific substructure. It is not general enough to represent arbitrary subgraphs, especially those with cycles due to the natural restriction of tree structure. Secondly, using rooted subtree as the elementary substructure results in a discriminating power bounded by the 1-WL test. For example, all $n$-node $r$-regular graphs cannot be discriminated by message passing GNNs. Thirdly, standard message passing GNNs do not allow using root-node-specific structural features (such as the distance between a node and the root node) to improve the quality of the learned root node's representation. We might need to break through such limitations in order to design more powerful GNNs.

## 3.2 The NGNN framework

To address the above limitations, we propose the Nested Graph Neural Network (NGNN) framework. NGNN no longer aims to encode a rooted subtree around each node. Instead, in NGNN, each node's final representation encodes the general local *subgraph* information around it more than a subtree, so that two graphs sharing a lot of identical or similar *rooted subgraphs* will have similar representations.

**Definition 1.** *(Rooted subgraph)* Given a graph $G$ and a node $v$, the height-$h$ rooted subgraph $G_v^h$ of $v$ is the subgraph induced from $G$ by the nodes within $h$ hops of $v$ (including $h$-hop nodes).

To make a node's final representation encode a rooted subgraph, we need to compute a subgraph representation. To achieve this, we resort to an arbitrary GNN, which we call the *base GNN* of NGNN. For example, the base GNN can be simply a message passing GNN, which performs message passing **within** each rooted subgraph to learn an intermediate representation for **every** node of the subgraph, and then uses a pooling layer to summarize a subgraph representation from the intermediate node representations. This subgraph representation is used as the final representation of the root node in

the original graph. Take root node $w$ as an example. We first perform $T$ rounds of message passing within node $w$'s rooted subgraph $G_w^h$. Let $v$ be any node appearing in $G_w^h$. We have

$$\boldsymbol{h}_{v,G_w^h}^{t+1} = U_t(\boldsymbol{h}_{v,G_w^h}^t, \boldsymbol{m}_{v,G_w^h}^{t+1}), \quad \text{where} \quad \boldsymbol{m}_{v,G_w^h}^{t+1} = \sum_{u \in N(v|G_w^h)} M_t(\boldsymbol{h}_{v,G_w^h}^t, \boldsymbol{h}_{u,G_w^h}^t, \boldsymbol{e}_{vu}). \quad (3)$$

Here $M_t, U_t$ are the message and update functions of the base GNN at time stamp $t$, $N(v|G_w^h)$ denotes the set of $v$'s neighbors within $w$'s rooted subgraph $G_w^h$, and $\boldsymbol{h}_{v,G_w^h}^{t+1}$ and $\boldsymbol{m}_{v,G_w^h}^{t+1}$ denote node $v$'s hidden state and message **specific to** rooted subgraph $G_w^h$ at time stamp $t+1$. Note that when node $v$ attends different nodes' rooted subgraphs, its hidden states and messages will also be **different**. This is in contrast to standard GNNs where a node's hidden state and message at time $t$ is the same regardless of which root node it contributes to. For example, $\boldsymbol{h}_v^{t+1}$ and $\boldsymbol{m}_v^{t+1}$ in Eq. 1 do not depend on any particular rooted subgraph.

After $T$ rounds of message passing, we apply a *subgraph pooling layer* to summarize a subgraph representation $\boldsymbol{h}_{G_w^h}$ from the intermediate node representations $\{\boldsymbol{h}_{v,G_w^h}^T | v \in G_w^h\}$.

$$\boldsymbol{h}_w := \boldsymbol{h}_{G_w^h} = R_0(\{\boldsymbol{h}_{v,G_w^h}^T | v \in G_w^h\}), \quad (4)$$

where $R_0$ is the subgraph pooling layer. This subgraph representation $\boldsymbol{h}_{G_w^h}$ will be used as root node $w$'s final representation $\boldsymbol{h}_w$ in the original graph. Note that the base GNNs are simultaneously applied to all nodes' rooted subgraphs to return a final node representation for every node in the original graph, and all the base GNNs share the same parameters. With such node representations, NGNN uses an *outer GNN* to further process and aggregate them into a graph representation of the whole graph. For simplicity, we let the outer GNN be simply a *graph pooling layer* denoted by $R_1$:

$$\boldsymbol{h}_G := R_1(\{\boldsymbol{h}_w | w \in G\}). \quad (5)$$

The Nested GNN framework can be understood as a two-level GNN, or a **GNN of GNNs**—the inner subgraph-level GNNs (base GNNs) are used to learn node representations from their rooted subgraphs, while the outer graph-level GNN is used to return a whole-graph representation from the inner GNNs' outputs. The inner GNNs all share the same parameters which are trained end-to-end with the outer GNN. Figure 2 depicts the implementation of the NGNN framework described above.

Compared to message passing GNNs, NGNN changes the "receptive field" of each node from a rooted subtree to a rooted subgraph, in order to capture better local substructure information. The rooted subgraph is read by a base GNN to learn a subgraph representation. Finally, the outer GNN reads the subgraph representations output by the base GNNs to return a graph representation.

Note that, when we apply the base GNN to a rooted subgraph, this rooted subgraph is extracted (copied) out of the original graph and treated as a completely independent graph from the other rooted subgraphs and the original graph. This allows the same node to have **different** representations within different rooted subgraphs. For example, in Figure 2, the same node $B$ appears in four different rooted subgraphs. Sometimes it is the root node, while other times it is a 1-hop neighbor of the root node. NGNN enables learning different representations for the same node when it appears in different rooted subgraphs, in contrast to standard GNNs where a node only has one single representation at one time stamp (Eq. 1). Similarly, NGNN also enables using different initial features for the same node when it appears in different rooted subgraphs. This allows us to customize a node's initial features based on its structural role within a rooted subgraph, as opposed to using the same initial features for a node across all rooted subgraphs. For example, we can optionally augment node $B$'s initial features with the distance between node $B$ and the root—when node $B$ is the root node, we give it an additional feature $0$; and when $B$ is a $k$-hop neighbor of the root, we give it an additional feature $k$. Such feature augmentation may help better capture a node's structural role within a rooted subgraph. It is an exclusive advantage of NGNN and is **not** possible in standard GNNs.

### 3.3 The representation power of NGNN

We theoretically characterize the additional expressive power of NGNN (using message passing GNNs as base GNNs) as opposed to standard message passing GNNs. We focus on the ability to discriminate regular graphs because they form an important category of graphs which standard GNNs cannot represent well. Using 1-WL or message passing GNNs, any two $n$-sized $r$-regular graphs will have the same representation, unless discriminative node features are available. In contrast, we prove that NGNN can distinguish almost all pairs of $n$-sized $r$-regular graphs regardless of node features.

**Definition 2.** *If the message passing (Eq. 3) and the two-level graph pooling (Eqs. 4,5) are all injective given input from a countable space, then the NGNN is called **proper**.*

A proper NGNN always exists due to the representation power of fully-connected neural networks used for message passing and Deep Set for graph pooling [28]. For all pairs of graphs that 1-WL can discriminate, there always exists a proper NGNN that can also discriminate them, because two graphs discriminated by 1-WL means they must have different multisets of rooted subtrees at some height $h$, while a rooted subtree is always included in a rooted subgraph with the same height.

Now we present our main theorem.

**Theorem 1.** *Consider all pairs of $n$-sized $r$-regular graphs, where $3 \leq r < (2 \log n)^{1/2}$. For any small constant $\epsilon > 0$, there exists a proper NGNN using at most $\lceil (\frac{1}{2} + \epsilon) \frac{\log n}{\log(r-1-\epsilon)} \rceil$-height rooted subgraphs and $\lceil \epsilon \frac{\log n}{\log(r-1-\epsilon)} \rceil$-layer message passing, which distinguishes almost all $(1 - o(1))$ such pairs of graphs.*

We include the proof in Appendix A. Theorem 1 has three implications. Firstly, since NGNN can discriminate almost all $r$-regular graphs where 1-WL always fails, it is **strictly more powerful** than 1-WL and message passing GNNs. Secondly, it implies that NGNN does not need to extract subgraphs with a too large height (about $\frac{1}{2} \frac{\log n}{\log(r-1)}$) to be more powerful. Moreover, NGNN is already powerful with very few layers, i.e., an arbitrarily small constant $\epsilon$ times $\frac{\log n}{\log(r-1)}$ (as few as 1 layer). This benefit comes from the subgraph pooling (Eq. 4), freeing us from using deep base GNNs. We further conduct a simulation experiment in Appendix D to verify Theorem 1 by testing how well NGNN discriminates $r$-regular graphs in practice. The results match almost perfectly with our theory.

Although NGNN is strictly more powerful than 1-WL and 2-WL (1-WL and 2-WL have the same discriminating power [20]), it is unclear whether NGNN is more powerful than 3-WL. Our early-stage analysis shows both NGNN and 3-WL cannot discriminate strongly regular graphs with the same parameters [29]. We leave the exact comparison between NGNN and 3-WL to future work.

### 3.4 Discussion

**Base GNN.** NGNN is a general plug-and-play framework to increase the power of a base GNN. For the base GNN, we are not restricted to message passing GNNs as described in Section 3.2. For example, we can also use GNNs approximating the power of higher-dimensional WL tests, such as 1-2-3-GNN [19] and PPGN/Ring-GNN [20, 21], as the base GNN. In fact, one limitation of these high-order GNNs is their $\mathcal{O}(n^3)$ complexity. Using the NGNN framework we can greatly alleviate this by applying the higher-order GNN to multiple small rooted subgraphs instead of the whole graph. Suppose a rooted subgraph has at most $c$ nodes, then by applying a high-order GNN to all $n$ rooted subgraphs, we can reduce the time complexity from $\mathcal{O}(n^3)$ to $\mathcal{O}(nc^3)$.

**Complexity.** We compare the time complexity of NGNN (using message passing GNNs as base GNNs) with a standard message passing GNN. Suppose the graph has $n$ nodes with a maximum degree $d$, and the maximum number of nodes in a rooted subgraph is $c$. Each message passing iteration in a standard message passing GNN takes $\mathcal{O}(nd)$ operations. In NGNN, we need to perform message passing over all $n$ nodes' rooted subgraphs, which takes $\mathcal{O}(n \cdot cd)$. We will keep $c$ small (which can be achieved by using a small $h$) to improve NGNN's scalability. Additionally, a small $c$ enables the base GNN to focus on learning local subgraph patterns.

In Appendix B, we discuss some other design choices of NGNN.

## 4 Related work

Understanding GNN's representation power is a fundamental problem in GNN research. Xu et al. [27] and Morris et al. [19] first proved that the discriminating power of message passing GNNs is bounded by the 1-WL test, namely they cannot discriminate two non-isomorphic graphs that 1-WL fails to discriminate (such as $r$-regular graphs). Since then, there is increasing effort in enhancing GNN's discriminating power beyond 1-WL [19, 21, 20, 30, 24, 31–33, 25]. Many GNNs have been proposed to mimic higher-dimensional WL tests, such as 1-2-3-GNN [19], Ring-GNN [21] and PPGN [20]. However, these models generally require learning the representations of all node tuples

of certain cardinality (e.g., node pairs, node triples and so on), thus cannot leverage the sparsity of graph structure and are difficult to scale to large graphs. Some works study the universality of GNNs for approximating any invariant or equivariant functions over graphs [34, 21, 35–37]. However, reaching universality would require polynomial($n$)-order tensors, which hold more theoretical value than practical applicability. Dasoulas et al. [38] propose to augment nodes of identical attributes with different colors, which requires exhausting all the coloring choices to reach universality. Similarly, Relational Pooling (RP) [30] uses the ensemble of permutation-aware functions over graphs to reach universality, which requires exhausting all $n!$ permutations to achieve its theoretical power. Its local version Local Relational Pooling (LRP) [39] applies RP over subgraphs around nodes, which is similar to our work yet still requires exhausting node permutations in local subgraphs and even more loses RP's theoretical power. In contrast, NGNN maintains a controllable cost by only applying a message passing GNN to local subgraphs, and is guaranteed to be more powerful than 1-WL.

Because of the high cost of mimicking high-dimensional WL tests, several works have been proposed to increase GNN's representation power within the message passing framework. Observing that different neighbors are indistinguishable during neighbor aggregation, some works propose to add one-hot node index features or random features to GNNs [40, 41]. These methods work well when nodes naturally have distinct identities irrespective of the graph structure. However, although making GNNs more discriminative, they also lose some of GNNs' generalization ability by not being able to guarantee nodes with identical neighborhoods to have the same embedding; the resulting models are also no longer permutation invariant. Repeating random initialization helps with avoiding such an issue but gets much slower convergence [42]. An exception is structural message-passing (SMP) [43], which propagates one-hot node index features to learn a global $n \times d$ feature matrix for each node. The feature matrix is further pooled to learn a permutation-invariant node representation.

On the contrary, some works propose to use structural features to augment GNNs without hurting the generalization ability of GNNs. SEAL [44, 45], IGMC [46] and DE [24] use distance-based features, where a distance vector w.r.t. the target node set to predict is calculated for each node as its additional features. Our NGNN framework is naturally compatible with such distance-based features due to its independent rooted subgraph processing. GSN [31] uses the count of certain substructures to augment node/edge features, which also surpasses 1-WL theoretically. However, GSN needs a properly defined substructure set to incorporate domain-specific inductive biases, while NGNN aims to learn arbitrary substructures around nodes without the need to predefine a substructure set.

Concurrent to our work, You et al. [32] propose Identity-aware GNN (ID-GNN). ID-GNN uses different weight parameters between each root node and its context nodes during message passing. It also extracts a rooted subgraph around each node, and thus can be viewed as a special case of NGNN with: 1) the number of message passing layers equivalent to the subgraph height, 2) directly using the root node's intermediate representation as its final representation without subgraph pooling, and 3) augmenting initial node features with 0/1 "identity". However, the extra power of ID-GNN only comes from the "identity" feature, while the power of NGNN comes from the subgraph pooling—without using any node features, NGNN is still provably more discriminative than 1-WL. Another similar work to ours is natural graph network (NGN) [47]. NGN argues that graph convolution weights need not be shared among all nodes but only (locally) isomorphic nodes. If we view our distance-based node features as refining the graph convolution weights so that nodes within a center node's neighborhood are no longer treated symmetrically, then our NGNN reduces to an NGN.

The idea of independently performing message passing within $k$-hop neighborhood is also explored in $k$-hop GNN [48] and MixHop [49]. However, MixHop directly concatenates the aggregation results of neighbors at different hops as the root representation, which ignores the connections between other nodes in the rooted subgraph. $k$-hop GNN sequentially performs message passing for $k$-hop, $k-1$-hop, ..., and 0-hop node (the update of $(i-1)$-hop nodes depend on the updated states of $i$-hop nodes), while NGNN simultaneously performs message passing for all nodes in the subgraph thus is more parallelizable. Both MixHop and $k$-hop GNN directly use the root node's representation as its final node representation. In contrast, NGNN uses a subgraph pooling to summarize all node representations within the subgraph as the final root representation, which distinguishes NGNN from other $k$-hop models. As Theorem 1 shows, the subgraph pooling enables using a much smaller number of message passing layers $l$ (as small as 1) than the depth $k$ of the subgraph, while MixHop and $k$-hop GNN always require $l \geq k$. MixHop and $k$-hop GNN also do not have the strong theoretical power of NGNN to discriminate $r$-regular graphs. Like SEAL and $k$-hop GNN, G-Meta [50] is another work extracting subgraphs around nodes/links. It focuses specifically on a meta-learning setting.

Table 1: Statistics and evaluation metrics of the QM9 and OGB datasets.

| Dataset | #Graphs | Avg. #nodes | Avg. #edges | Split ratio | #Tasks | Task type | Metric |
|---------|---------|-------------|-------------|-------------|--------|-----------|--------|
| QM9 | 129,433 | 18.0 | 18.6 | 80/10/10 | 12 | Regression | MAE |
| ogbl-molhiv | 41,127 | 25.5 | 27.5 | 80/10/10 | 1 | Classification | ROC-AUC |
| ogbl-molpcba | 437,929 | 26.0 | 28.1 | 80/10/10 | 128 | Classification | AP |

## 5 Experiments

In this section, we study the effectiveness of the NGNN framework for graph classification and regression tasks. In particular, we want to answer the following questions:

**Q1** Can NGNN reach its theoretical power to discriminate 1-WL-indistinguishable graphs?
**Q2** How often and how much does NGNN improve the performance of a base GNN?
**Q3** How does NGNN perform in comparison to state-of-the-art GNN methods in open benchmarks?
**Q4** How much extra computation time does NGNN incur?

We implement the NGNN framework based on the PyTorch Geometric library [51]. Our code is available at `https://github.com/muhanzhang/NestedGNN`.

### 5.1 Datasets

To answer **Q1**, we use a simulation dataset of $r$-regular graphs and the EXP dataset [42] containing 600 pairs of 1-WL-indistinguishable but non-isomorphic graphs. To answer **Q2**, we use the QM9 dataset [52, 53] and the TU datasets [54]. QM9 contains 130K small molecules. The task here is to perform regression on twelve targets representing energetic, electronic, geometric, and thermodynamic properties, based on the graph structure and node/edge features. TU contains five graph classification datasets including D&D [55], MUTAG [56], PROTEINS [55], PTC_MR [57], and ENZYMES [58]. We used the datasets provided by PyTorch Geometric [51], where for QM9 we performed unit conversions to match the units used by [19]. The evaluation metric is Mean Absolute Error (MAE) for QM9 and Accuracy (%) for TU. To answer **Q3**, we use two Open Graph Benchmark (OGB) datasets [59], `ogbg-molhiv` and `ogbg-molpcba`. The `ogbg-molhiv` dataset contains 41K small molecules, the task of which is to classify whether a molecule inhibits HIV virus or not. ROC-AUC is used for evaluation. The `ogbg-molpcba` dataset contains 438K molecules with 128 classification tasks. The evaluation metric is Average Precision (AP) averaged over all the tasks. We include the statistics for QM9 and OGB datasets in Table 1.

### 5.2 Models

**QM9.** We use 1-GNN, 1-2-GNN, 1-3-GNN, and 1-2-3-GNN from [19] as both the baselines and the base GNNs of NGNN. Among them, 1-GNN is a standard message passing GNN with 1-WL power. 1-2-GNN is a GNN mimicking 2-WL, where message passing happens among 2-tuples of nodes. 1-3-GNN and 1-2-3-GNN mimic 3-WL, where message passing happens among 3-tuples of nodes. 1-2-GNN and 1-3-GNN use features computed by 1-GNN as initial node features, and 1-2-3-GNN uses the concatenated features from 1-2-GNN and 1-3-GNN. We additionally include numbers provided by [53] and Deep LRP [39] as baselines. Note that we omit more recent methods [60–62] using advanced physical representations calculated from angles, atom coordinates, and quantum mechanics, which may obscure the comparison of models' pure graph representation power. For NGNN, we uniformly use height-3 rooted subgraphs. For a fair comparison, the base GNNs in NGNN use exactly the same hyperparameters as when they are used alone, except for 1-GNN where we increase the number of message passing layers from 3 to 5 to make the number of layers larger than the subgraph height, similar to [63]. For subgraph pooling and graph pooling layers, we uniformly use mean pooling. All other settings follow [19].

**TU.** We use four widely adopted GNNs as the baselines and the base GNNs of NGNN: GCN [12], GraphSAGE [64], GIN [27], and GAT [15]. Since TU datasets suffer from inconsistent evaluation standards [65], we uniformly use the 10-fold cross validation framework provided by PyTorch Geomtric [66] for all the models to ensure a fair comparison. For GNNs, we search the number of message passing layers in $\{2, 3, 4, 5\}$. For NGNNs, we similarly search the subgraph height $h$ in $\{2, 3, 4, 5\}$, so that both NGNNs and GNNs can have equal-depth local receptive fields. For NGNNs, we always use $h + 1$ message passing layers instead of searching it together with $h$, because that will

make NGNNs have more hyperparameters to tune. All models have 32 hidden dimensions, and are trained for 100 epochs with a batch size of 128. For each fold, we record the test accuracy with the hyperparameters chosen based on the best validation performance of this fold. Finally, we report the average test accuracy across all the 10 folds.

**OGB.** We use GNNs achieving top places on the OGB graph classification leaderboard[3] (at the time of submission) as the baselines, including GCN [12], GIN [27], DeeperGCN [67], Deep LRP [39], PNA [68], DGN [33], GINE [69], and PHC-GNN [70]. Note that those high-order GNNs [19–21, 25] are not included here, because despite being theoretically more discriminative, these GNNs are **not** among the GNNs with the best empirical performance on modern large-scale graph benchmarks, and their $\mathcal{O}(n^3)$ complexity also raises a scalability issue. For NGNN, we use GIN as the base GNN (although GIN is not among the strongest baselines here). Some baselines additionally use the virtual node technique [18, 11, 71], which are marked by "*". For NGNN, we search the subgraph height $h$ in $\{3, 4, 5\}$, and the number of layers in $\{4, 5, 6\}$. We train the NGNN models for 100 and 150 epochs for `ogbg-molhiv` and `ogbg-molpcba`, respectively, and report the validation and test scores at the best validation epoch. We also find that our models are subject to high performance variance across epochs, likely due to the increased expressiveness. Thus, we save a model checkpoint every 10 epochs, and additionally report the ensemble performance by averaging the predictions from all checkpoints. The final hyperparameter choices and more details about the experimental settings are included in Appendix C. All results are averaged over 10 independent runs.

In the following, we uniformly use "Nested GNN" to denote an NGNN model using "GNN" as the base GNN. For example, Nested GIN denotes an NGNN model using GIN [27] as the base GNN. For the NGNN models in QM9, TU and OGB datasets, we augment the initial features of a node with Distance Encoding (DE) [24], which uses the (generalized) distance between a node and the root as its additional feature, due to DE's successful applications in link-level tasks [44, 46]. Note that such feature augmentation is not applicable to the baseline models as discussed in Section 3.2. An ablation study on the effects of the DE features is included in Appendix E.

### 5.3 Results and discussion

To answer **Q1**, we first run a simulation to test NGNN's power for discriminating $r$-regular graphs. The results are presented in Appendix D. They match almost perfectly with Theorem 1, demonstrating that a practical NGNN can fulfil its theoretical power for discriminating $r$-regular graphs.

Table 2: Results (%) on EXP.

| Method | Test Accuracy |
|---|---|
| **GCN-RNI** [42] | 98.0±1.85 |
| **PPGN** [20] | 50.0±0.00 |
| **1-2-3-GNN** [19] | 50.0±0.00 |
| **3-GCN** [42] | 99.7±0.004 |
| **Nested GIN** | **99.9±0.26** |

We also test NGNN's expressive power using the EXP dataset provided by [42], which contains 600 carefully constructed 1-WL indistinguishable but non-isomorphic graph pairs. Each pair of graphs have different labels, thus a standard message passing GNN cannot predict them both correctly, resulting in an expected classification accuracy of only 50%. We exactly follow the experimental settings and copy the baseline results in [42]. In Table 2, our Nested GIN model achieves a 99.9% classification accuracy, which outperforms all the baselines and distinguishes almost all the 1-WL indistinguishable graph pairs. These results verified that NGNN's expressive power is indeed beyond 1-WL and message passing GNNs.

To answer **Q2**, we adopt the QM9 and TU datasets. We show the QM9 results in Table 3. If the Nested version of a base GNN achieves a better result than the base GNN itself, we color that cell with light green. As we can see, NGNN brings performance gains to all base GNNs on most targets, sometimes by large margins. We also show the results on TU in Table 4. NGNNs also show improvement over their base GNNs in most cases. These results indicate that NGNN is a general framework for improving a GNN's power. We further compute the maximum reduction of MAE for QM9 and maximum improvement of accuracy for TU before and after applying NGNN. NGNN reduces the MAE by up to 7.9 times for QM9, and increases the accuracy by up to 14.3% for TU. These results answer **Q2**, indicating that NGNN can bring steady and significant improvement to base GNNs.

To answer **Q3**, we compare Nested GIN with leading methods on the OGB leaderboard. The results are shown in Table 5. Nested GIN achieves highly competitive performance with these leading GNN models, albeit using a relatively weak base GNN (GIN). Compared to GIN alone, Nested GIN shows clear performance gains. It achieves test scores up to 79.86 and 30.07 on `ogbg-molhiv` and

---

[3]`https://ogb.stanford.edu/docs/leader_graphprop/`

Table 3: MAE results on QM9 (smaller the better). A colored cell means NGNN is better than the base GNN.

| Target | Method (Ne. for Nested) | | | | | | | | | | | |
|---|---|---|---|---|---|---|---|---|---|---|---|---|
| | DTNN | MPNN | Deep LRP | 1-GNN | 1-2-GNN | 1-3-GNN | 1-2-3-GNN | Ne. 1-GNN | Ne. 1-2-GNN | Ne. 1-3-GNN | Ne. 1-2-3-GNN | Max. reduction |
| $\mu$ | **0.244** | 0.358 | 0.364 | 0.493 | 0.493 | 0.473 | 0.476 | 0.428 | 0.437 | 0.436 | 0.433 | 1.2× |
| $\alpha$ | 0.95 | 0.89 | 0.298 | 0.78 | 0.27 | 0.46 | 0.27 | 0.29 | 0.278 | **0.261** | 0.265 | 2.7× |
| $\varepsilon_{\text{HOMO}}$ | 0.00388 | 0.00541 | **0.00254** | 0.00321 | 0.00331 | 0.00328 | 0.00337 | 0.00265 | 0.00275 | 0.00265 | 0.00279 | 1.2× |
| $\varepsilon_{\text{LUMO}}$ | 0.00512 | 0.00623 | 0.00277 | 0.00355 | 0.00350 | 0.00354 | 0.00351 | 0.00297 | 0.00271 | **0.00269** | 0.00276 | 1.3× |
| $\Delta\varepsilon$ | 0.0112 | 0.0066 | **0.00353** | 0.0049 | 0.0047 | 0.0046 | 0.0048 | 0.0038 | 0.0039 | 0.0039 | 0.0039 | 1.8× |
| $\langle R^2 \rangle$ | **17.0** | 28.5 | 19.3 | 34.1 | 21.5 | 25.8 | 22.9 | 20.5 | 20.4 | 20.2 | 20.1 | 1.7× |
| ZPVE | 0.00172 | 0.00216 | 0.00055 | 0.00124 | 0.00018 | 0.00064 | 0.00019 | 0.00020 | 0.00017 | 0.00017 | **0.00015** | 6.2× |
| $U_0$ | 2.43 | 2.05 | 0.413 | 2.32 | **0.0357** | 0.6855 | 0.0427 | 0.295 | 0.252 | 0.291 | 0.205 | 7.9× |
| $U$ | 2.43 | 2.00 | 0.413 | 2.08 | **0.107** | 0.686 | 0.111 | 0.361 | 0.265 | 0.278 | 0.200 | 5.8× |
| $H$ | 2.43 | 2.02 | 0.413 | 2.23 | 0.070 | 0.794 | **0.0419** | 0.305 | 0.241 | 0.267 | 0.249 | 7.3× |
| $G$ | 2.43 | 2.02 | 0.413 | 1.94 | 0.140 | 0.587 | **0.0469** | 0.489 | 0.272 | 0.287 | 0.253 | 4.0× |
| $C_v$ | 0.27 | 0.42 | 0.129 | 0.27 | 0.0989 | 0.158 | 0.0944 | 0.174 | 0.0891 | 0.0879 | **0.0811** | 1.8× |

Table 4: Accuracy results (%) on TU datasets.

| | D&D | MUTAG | PROTEINS | PTC_MR | ENZYMES |
|---|---|---|---|---|---|
| #Graphs | 1178 | 188 | 1113 | 344 | 600 |
| Avg. #nodes | 284.32 | 17.93 | 39.06 | 14.29 | 32.63 |
| GCN | 71.6±2.8 | 73.4±10.8 | 71.7±4.7 | 56.4±7.1 | 27.3±5.5 |
| GraphSAGE | 71.6±3.0 | 74.0±8.8 | 71.2±5.2 | 57.0±5.5 | 30.7±6.3 |
| GIN | 70.5±3.9 | 84.5±8.9 | 70.6±4.3 | 51.2±9.2 | **38.3**±6.4 |
| GAT | 71.0±4.4 | 73.9±10.7 | 72.0±3.3 | 57.0±7.3 | 30.2±4.2 |
| Nested GCN | 76.3±3.8 | 82.9±11.1 | 73.3±4.0 | **57.3**±7.7 | 31.2±6.7 |
| Nested GraphSAGE | 77.4±4.2 | 83.9±10.7 | **74.2**±3.7 | 57.0±5.9 | 30.7±6.3 |
| Nested GIN | **77.8**±3.9 | **87.9**±8.2 | 73.9±5.1 | 54.1±7.7 | 29.0±8.0 |
| Nested GAT | 76.0±4.4 | 81.9±10.2 | 73.7±4.8 | 56.7±8.1 | 29.5±5.7 |
| Max. improvement | 10.4% | 13.4% | 4.7% | 5.7% | 14.3% |

Table 5: Results (%) on OGB datasets (* virtual node).

| | ogbg-molhiv (AUC) | | ogbg-molpcba (AP) | |
|---|---|---|---|---|
| Method | Validation | Test | Validation | Test |
| CCN* | 83.84±0.91 | 75.99±1.19 | 24.95±0.42 | 24.24±0.34 |
| GIN* | 84.79±0.68 | 77.07±1.49 | 27.98±0.25 | 27.03±0.23 |
| Deep LRP | 82.09±1.16 | 77.19±1.40 | – | – |
| DeeperGCN* | – | – | 29.20±0.25 | 27.81±0.38 |
| HIMP | – | 78.80±0.82 | – | – |
| PNA | 85.19±0.99 | 79.05±1.32 | – | – |
| DGN | 84.70±0.47 | 79.70±0.97 – | – | |
| GINE* | – | – | 30.65±0.30 | 29.17±0.15 |
| PHC-GNN | 82.17±0.89 | 79.34±1.16 | 30.68±0.25 | 29.47±0.26 |
| Nested GIN* | 83.17±1.99 | 78.34±1.86 | 29.15±0.35 | 28.32±0.41 |
| Nested GIN* (ens) | 80.80±2.78 | **79.86**±1.05 | 30.59±0.56 | **30.07**±0.37 |

`ogbg-molpcba`, respectively, which outperform all the baselines. In particular, for the challenging `ogbg-molpcba`, our Nested GIN can achieve 30.07 and 28.32 test AP with and without ensemble, respectively, outperforming the plain GIN model (with 27.03 test AP) significantly. These results demonstrate the great empirical performance and potential of NGNN even compared to heavily tuned open leaderboard models, despite using only GIN as the base GNN.

To answer **Q4**, we report the training time per epoch for GIN and Nested GIN on OGB datasets. On `ogbg-molhiv`, GIN takes 54s per epoch, while Nested GIN takes 183s. On `ogbg-molpcba`, GIN takes 10min per epoch, while Nested GIN takes 20min. This verifies that NGNN has comparable time complexity with message passing GNNs. The extra complexity comes from independently learning better node representations from rooted subgraphs, which is a trade-off for the higher expressivity.

In summary, our experiments have firmly shown that NGNN is a theoretically sound method which brings consistent gains to its base GNNs in a plug-and-play way. Furthermore, NGNN still maintains a controllable time complexity compared to other more powerful GNNs.

Finally, we point out one memory limitation of the current NGNN implementation. Currently, NGNN does not scale to graph datasets with a large average node number (such as REDDIT-BINARY) or datasets with a large average node degree (such as `ogbg-ppa`) due to copying a rooted subgraph for each node to the GPU memory. Reducing batch size or subgraph height helps, but at the same time leads to performance degradation. One may wonder why materializing all the subgraphs into GPU memory is necessary. The reason is that we want to batch-process all the subgraphs simultaneously. Otherwise, we have to sequentially extract subgraphs on the fly, which results in a much higher latency. We leave the exploration of memory efficient NGNN to the future work.

# 6 Conclusions

We have proposed Nested Graph Neural Network (NGNN), a general framework for improving GNN's representation power. NGNN learns node representations encoding rooted subgraphs instead of rooted subtrees. Theoretically, we prove NGNN can discriminate almost all $r$-regular graphs where 1-WL always fails. Empirically, NGNN consistently improves the performance of various base GNNs across different datasets without incurring the $\mathcal{O}(n^3)$ complexity like other more powerful GNNs.

# Acknowledge

The authors greatly thank the actionable suggestions from the reviewers to improve the manuscript. Li is partly supported by the 2021 JP Morgan Faculty Award and the National Science Foundation (NSF) award HDR-2117997.

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
