## A  Proof of Theorem 1

The proof is inspired by the previous theoretical characterization on the power of distance features [24]. Basically, performing height-$k$ subgraph extraction around a center node is essentially equivalent to injecting distance features that indicate whether the distance between a node and the center node is less than $k + 1$. In the following part, we will explicitly show how these distance features make NGNN more powerful than the 1-WL test. Let us first introduce the outline of the proof. Consider two $n$-node $r$-regular graphs $G^{(1)} = (V^{(1)}, E^{(1)})$ and $G^{(2)} = (V^{(2)}, E^{(2)})$ and we pick two nodes, each from one graph, denoted by $v_1$ and $v_2$. By performing certain-height (at most $\lceil (\frac{1}{2} + \epsilon) \frac{\log n}{\log(r-1)} \rceil$-height) rooted subgraph extraction around these two nodes, due to the implicit distance features, we may prove that the nodes on the boundary of the obtained two subgraphs will obtain special node representations. These special node representations will be propagated within the subgraphs. After some steps of propagation, we can prove that NGNN by leveraging the subgraph pooling (Eq. 4) can distinguish these two subgraphs. This tells that NGNN may generate different node representations

for $v_1$ and $v_2$ respectively. Then, a union bound can be used to transform such difference in node representations into the difference in the representations of $G^{(1)}$ and $G^{(2)}$. Note that the proof will assume that there are no node/edge attributes that can be leveraged. Additional node/edge attributes may only improve the possibility to distinguish these two graphs.

The first lemma is to analyze the difference between the structures of the rooted subgraphs around two nodes over two $n$-node $r$-regular graphs. Before introducing that, we need to define a notion termed edge configuration. For a node $v$ in graph $G$, let $Q_{v,G}^k$ denote the set of nodes in $G$ that are exactly $k$-hop neighbors of $v$, i.e., the shortest path distance between $v$ and any node $u \in Q_{v,G}^k$ is $k$. Then, we know the height-$k$ rooted subgraph over $G$ around the center node $v$ is the subgraph induced by the node set $\cup_{i=0}^k Q_{v,G}^i$.

**Definition 3.** *The edge configuration between $Q_{v,G}^k$ and $Q_{v,G}^{k+1}$ is a list $C_{v,G}^k = (a_{v,G}^{1,k}, a_{v,G}^{2,k}, ...)$ where $a_{v,G}^{i,k}$ denotes the number of nodes in $Q_{v,G}^{k+1}$ of which each has exactly $i$ edges from $Q_{v,G}^k$.*

When we say two edge configurations $C_{v_1,G^{(1)}}^k$ (between $Q_{v_1,G^{(1)}}^k$ and $Q_{v_1,G^{(1)}}^{k+1}$), $C_{v_2,G^{(2)}}^k$ (between $Q_{v_2,G^{(2)}}^k$ and $Q_{v_2,G^{(2)}}^{k+1}$) are equal, we mean that these two lists are component-wise equal to each other. Obviously, we should also have $|Q_{v_1,G^{(1)}}^{k+1}| = |Q_{v_2,G^{(2)}}^{k+1}|$ if $C_{v_1,G^{(1)}}^k = C_{v_2,G^{(2)}}^k$. Now, we are ready to propose the first lemma.

**Lemma 1.** *For two graphs $G^{(1)} = (V^{(1)}, E^{(1)})$ and $G^{(2)} = (V^{(2)}, E^{(2)})$ that are uniformly independently sampled from all $n$-node $r$-regular graphs, where $3 \leq r < \sqrt{2\log n}$, we pick any two nodes, each from one graph, denoted by $v_1$ and $v_2$ respectively. Then, there is at least one $i \in (\frac{1}{2} \frac{\log n}{\log(r-1-\epsilon)}, (\frac{1}{2} + \epsilon) \frac{\log n}{\log(r-1-\epsilon)})$ with probability $1 - o(n^{-1})$ such that $C_{v_1,G^{(1)}}^i \neq C_{v_2,G^{(2)}}^i$. Moreover, with at least the same probability, for all $i \in (\frac{1}{2} \frac{\log n}{\log(r-1-\epsilon)}, (\frac{2}{3} - \epsilon) \frac{\log n}{\log(r-1)})$, the number of edges between $Q_{v_j,G^{(j)}}^i$ and $Q_{v_j,G^{(j)}}^{i+1}$ are at least $(r-1-\epsilon)|Q_{v_j,G^{(j)}}^i|$ for $j \in \{1, 2\}$.*

*Proof.* This lemma can be obtained by following the steps 1-3 in the proof of Theorem 3.3 in [24]. $\square$

Now, we set $K = \lceil (\frac{1}{2} + \epsilon) \frac{\log n}{\log(r-1-\epsilon)} \rceil$. We focus on the two extracted subgraphs $G_{v_1}^K$ and $G_{v_2}^K$. We first prove a lemma that shows with a certain number of layers, a proper NGNN will generate different representations for $G_{v_1}^K$ and $G_{v_2}^K$, i.e., $\boldsymbol{h}_{v_1}$ and $\boldsymbol{h}_{v_2}$ in Eq. 4.

**Lemma 2.** *For two graphs $G^{(1)} = (V^{(1)}, E^{(1)})$ and $G^{(2)} = (V^{(2)}, E^{(2)})$ that are uniformly independently sampled from all $n$-node $r$-regular graphs, where $3 \leq r < \sqrt{2\log n}$, we pick any two nodes, each from one graph, denoted by $v_1$ and $v_2$ respectively, and do $\lceil (\frac{1}{2} + \epsilon) \frac{\log n}{\log(r-1-\epsilon)} \rceil$-height rooted subgraph extraction around $v_1$ and $v_2$. With at most $\epsilon \lceil \frac{\log n}{\log(r-1-\epsilon)} \rceil$ many layers, a proper message passing GNN (with injective $U_t$, $M_t$ and subgraph pooling) will generate different representations for the extracted two subgraphs with probability at least $1 - o(n^{-1})$.*

*Proof.* According to Lemma 1, we know that with probability $1 - o(n^{-1})$, there exists at least one $i \in (\frac{1}{2} \frac{\log n}{\log(r-1-\epsilon)}, (\frac{1}{2} + \epsilon) \frac{\log n}{\log(r-1-\epsilon)})$ such that $C_{v_1,G^{(1)}}^k \neq C_{v_2,G^{(2)}}^k$. So there exists at least one $k \leq K$ that make $C_{v_1,G^{(1)}}^k \neq C_{v_2,G^{(2)}}^k$ (thus the difference in edge configurations appears in $G_{v_1}^K$ and $G_{v_2}^K$) and we pick the largest $k$.

Now let us consider running a message passing GNN over the two subgraphs $G_{v_j}^K$, $j \in \{1, 2\}$. All nodes are initialized with the same node features. The nodes of these two subgraphs can be categorized into $Q_{v_j,G^{(j)}}^i$ ($0 \leq i \leq K$), for $j \in \{1, 2\}$ respectively. Next, let us consider the node representations in these categories during the message passing procedure. We have the following observations.

1. Note that all the nodes other than those in $Q_{v_j,G^{(j)}}^K$ have degree $r$ in both subgraphs. Therefore, in the $t$-th iteration, the nodes in $\cup_{i=0}^{K-t} Q_{v_j,G^{(j)}}^i$ for $j \in \{1, 2\}$ will share the same node representation. We call this node representation as *default representation*. Note

that if we do not perform rooted subgraph extraction, then all nodes in all $r$-regular graph hold default representation.

2. Node representations that are different from default representations will first appear among the nodes in $Q^K_{v_j,G^{(j)}}$ after the first iteration. This is because there are at least $(r-1-\epsilon)|Q^K_{v_j,G^{(j)}}|$ edges between $Q^K_{v_j,G^{(j)}}$ and $Q^{K+1}_{v_j,G^{(j)}}$ before performing subgraph extraction (due to Lemma 1) and all these edges will not appear in the extracted subgraphs. Then, almost all nodes in $Q^K_{v_j,G^{(j)}}$ hold only degree one (and thus do not have degree $r$ to keep default representations) within the corresponding extracted subgraphs. We uniformly call the node representations that are different from the default ones as *new representations*. New representations may be mutually different.

3. Those new different node representations will propagate to nodes in $Q^{K-1}_{v_j,G^{(j)}}$, $Q^{K-2}_{v_j,G^{(j)}}$ and so on and so forth via iterative message passing. Moreover, during such propagation procedure, after $t \geq 2$ iterations, new representations will at least make almost all nodes in $Q^i_{v_j,G^{(j)}}$ hold representations different form almost all nodes in $Q^{i+1}_{v_j,G^{(j)}}$ for $i = K - t + 1, K - t + 2, ..., K - 1$, which can be easily obtained by doing induction from $t = t_1$ to $t = t_1 + 1$.

Observing the above three points, We may compare the above propagating procedure between $G^K_{v_1}$ and $G^K_{v_2}$. Suppose in the first $K - k$ steps of message passing, the set of node representations (both the default ones and the new ones) can keep the same between the two extracted subgraphs. If this is not true, we have already proven the results. As they hold different edge configurations in $C^k_{v_1,G^{(1)}} \neq C^k_{v_2,G^{(2)}}$, when the new node representations propagate from $Q^{k+1}_{v_j,G^{(j)}}$ to $Q^k_{v_j,G^{(j)}}$, it will definitely induce different sets of new node representations between $Q^k_{v_1,G^{(1)}}$ and $Q^k_{v_2,G^{(2)}}$. Currently, node representations are kept the same between $Q^i_{v_1,G^{(1)}}$ and $Q^i_{v_2,G^{(2)}}$ for $i \neq [0, k-1]$ as they are all default node representations. Though $\cup^K_{i=k+1}Q^i_{v_j,G^{(j)}}$ also hold new node representations, they are different from those in $Q^k_{v_j,G^{(j)}}$ for $j \in \{1, 2\}$. At this point, if an injective subgraph pooling operation is adopted, then the obtained representations of $G^K_{v_1}$ and $G^K_{v_2}$, i.e., $\boldsymbol{h}_{v_1}$ and $\boldsymbol{h}_{v_2}$, are different. □

Based on Lemma 2, using a union bound by comparing a node representation of $G^{(1)}$ with all node representaitons of $G^{(2)}$, we may achieve the final conclusion. Specifically, we consider a node of $G^{(1)}$, say $v_1$, and another arbitrary node of $G^{(2)}$, say $v_2$. Using Lemma 2, we know with probability $1 - o(n^{-1})$, $\boldsymbol{h}_{v_1}$ is different from $\boldsymbol{h}_{v_2}$. Then, using the union bound, with probability $1 - o(1)$, we have $\boldsymbol{h}_{v_1} \notin \{\boldsymbol{h}_{v_2}|v_2 \in V(G^{(2)})\}$. Therefore, if the final graph pooling (Eq. 5) is injective, we may guarantee that NGNN can generate different representations for $G^{(1)}$ and $G^{(2)}$.

## B  Design choices of NGNN

In this section, we discuss some other design choices of NGNN.

**High-order NGNN.** NGNN is a two-level GNN (a GNN of GNNs), where a base GNN is used to learn a final node representation from a rooted subgraph and an outer GNN (graph pooling) is used to learn a graph representation from the base GNNs' outputs. This design thus involves one level of nesting, which we call first-order NGNN. To extend the framework, we propose *high-order NGNN*, where we make the base GNN itself an NGNN. That is, we perform the subgraph representation learning tasks each using a first-order NGNN, where we treat each subgraph the same as the graph in the original NGNN. This way, we arrive at a second-order NGNN with two levels of nesting (a GNN of NGNNs, or a GNN of GNNs of GNNs). Repeating this construction, we can in principle construct an arbitrary-order NGNN. It is interesting to investigate whether high-order NGNNs can further enhance the representation power and the practical performance of a base GNN. We leave the exploration of such architectures to future work.

**Pooling functions $R_0$ and $R_1$.** To summarize node representations into a subgraph/graph representation, we need a readout (pooling) function. Popular choices include sum, mean, max, as well as

more complex ones such as selecting top-$K$ nodes [16, 72] and hierarchical approaches [17]. In this paper, we find mean pooling works very well, which directly takes the mean of node representations as the subgraph/graph representation. We also find another pooling function to be sometimes useful for subgraph pooling, called center pooling (CP). CP directly uses the root node's representation to represent the entire subgraph. The success of CP relies on using more layers of message passing than the height of the rooted subgraph, so that even the intermediate representation of the center root node alone can have sufficient information about the entire subgraph. This is feasible for rooted subgraphs with a small height. Note that when using a number of message passing layers smaller than the subgraph height, NGNN with CP will reduce to a standard message passing GNN.

**Subgraph height $h$ and base GNN layers $l$.** NGNN is flexible in terms of choosing the subgraph height $h$ and the number of message passing layers $l$ in the base GNN. Theorem 1 provides a guide for choosing $h$ and $l$ when discriminating $r$-regular graphs. In practice, we find using $h = 3$ and $l = 4$ generally performs well across various tasks. Using a small $h$ will restrict the receptive field, causing NGNN to learn too local features. Using a too large $h$ might cause each rooted subgraph to include the entire graph. For the number of message passing layers $l$, we find that using $l \geq h$ performs better. This can be explained by that using a large $l$ makes each node in a rooted subgraph to more sufficiently absorb the whole-subgraph information thus learning a better intermediate node representation reflecting its structural position within the subgraph. Please refer to [63] for more motivations for using deeper message passing layers than the subgraph height.

## C    More details about the experimental settings

The experiments were run on a Linux server with 64GB memory, two NVIDIA RTX 2080S (8GB) GPUs and an INTEL i9-9900 8-core CPU. For `ogbg-molhiv`, the final NGNN architecture used a rooted subgraph height $h = 4$ and number of GIN layers $l = 6$. Mean pooling is used in both the subgraph and graph pooling. The final NGNN architecture for `ogbg-molpcba` used a rooted subgraph height $h = 3$ and the number of GIN layers $l = 4$. Center pooling (CP) is used in the subgraph pooling and mean pooling is used in the graph pooling. Although we searched $h$ and $l$, we found the final performance is not very sensitive to these hyperparameters as long as $h$ is between 3 and 5 and $l > h$. For the DE features, we use shortest path distance and resistance distance [73].

## D    Simulation experiments to verify Theorem 1

We conduct a simulation over random regular graphs to validate Lemma 2 (how well NGNN distinguishes nodes of regular graphs) and Theorem 1 (how well NGNN distinguishes regular graphs). The results are shown in Figure 3, which match our theory almost perfectly. Basically, we sample 100 $n$-node 3-regular graphs uniformly at random, and then apply an untrained NGNN to these graphs to see how often NGNN can distinguish the nodes and graphs at different rooted subgraph height $h$ and node number $n$. The required $h$ at different $n$ matches almost perfectly with the lower bound in Lemma 2. More details are contained in the caption of Figure 3.

## E    Ablation study on DE

In this paper, we choose Distance Encoding (DE) [24] to augment the initial node features of NGNN, due to its good theoretical properties for improving the expressive power of message passing GNNs as well as its superb empirical performance on link prediction tasks [44, 46]. DE encodes the distance between a node and the root node into a vector through an embedding layer. The distance embedding is concatenated with the raw features of a node as its new features (in this rooted subgraph) input to the base GNN. Note that when this node appears in another rooted subgraph, it may have a different distance to that root node, thus resulting in different DE features in different subgraphs. Only the NGNN framework can leverage such a subgraph-specific feature augmentation—a standard GNN treats a node always the same no matter which node's rooted subgraph/subtree it is in.

In this section, we do ablation experiments to study the effect of the DE features. We choose QM9 as the testbed. The base GNNs are the same as in Table 3. For each base GNN, we compare it with its Nested GNN version without DE features (no DE) and its Nested GNN version with DE features (with DE). The results are shown in Table 6.

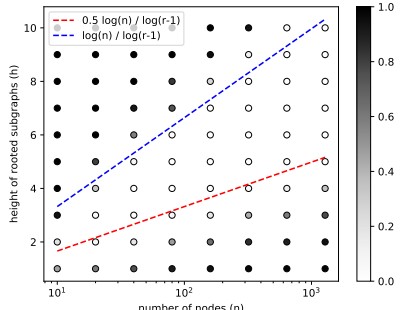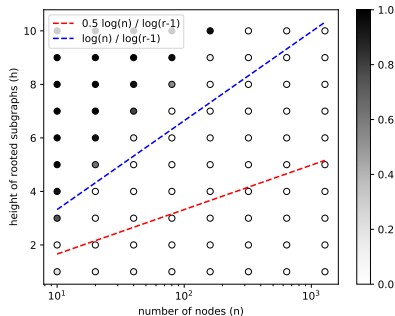

Figure 3: Simulation to verify Theorem 1. The left graph shows the node-level (with only subgraph pooling) simulation results. The right graph shows the graph-level (with both subgraph and graph pooling) simulation results. We uniformly sample 100 $n$-node 3-regular graphs with $n$ ranging from 10 to 1280. We let the rooted subgraph height $h$ range from 1 to 10. We apply an untrained Nested GIN with one message passing layer to these graphs (with a uniform 1 as node features). In the left figure, we compare the final node representations (after subgraph pooling) from all graphs output by the Nested GIN. If the difference between two node representations $||\boldsymbol{h}_u - \boldsymbol{h}_v||_2$ is greater than machine accuracy, they are regarded as indistinguishable. The shade of each scatter point's color reflects the portion of indistinguishable node pairs at certain $(n, h)$. The darker, the more indistinguishable node pairs. In the right graph, we compare the final graph representations (after graph pooling) output by the Nested GIN. The blue and red dashed lines show the theoretical upper and lower bounds for $h$ to discriminate almost all nodes in $n$-node 3-regular graphs, respectively. As we can see, the node-level simulation results perfectly match the theory (Lemma 2)—when $h$ is larger than $0.5 \log(n)/\log(r-1)$, almost all nodes from $r$-regular graphs are distinguishable by NGNN. When $h$ is even larger than $\log(n)/\log(r-1)$, the nodes can hardly be distinguished because each subgraph contains the entire regular graph. The graph-level simulation results show that even using a very small $h$ NGNN can still discriminate almost all $r$-regular graphs—$h$ in practice even does not need to be always chosen beyond $0.5 \log(n)/\log(r-1)$. This is because although most nodes from two $r$-regular graphs cannot be distinguished when $h \leq 0.5 \log(n)/\log(r-1)$, the graph pooling can still distinguish the two graphs as long as there exists one single node from one graph holding a representation different from any node representation from the other graph.

Table 6: Ablation study on QM9 comparing Nested GNNs with and without DE features.

| Method | $\mu$ | $\alpha$ | $\varepsilon_{\text{HOMO}}$ | $\varepsilon_{\text{LUMO}}$ | $\Delta\varepsilon$ | $\langle R^2 \rangle$ | ZPVE | $U_0$ | $U$ | $H$ | $G$ | $C_v$ |
|---|---|---|---|---|---|---|---|---|---|---|---|---|
| **1-GNN** | 0.493 | 0.78 | 0.00321 | 0.00355 | 0.0049 | 34.1 | 0.00124 | 2.32 | 2.08 | 2.23 | 1.94 | 0.27 |
| **Nested 1-GNN** (no DE) | 0.466 | 0.38 | 0.00292 | **0.00294** | 0.0042 | 24.0 | 0.00040 | 1.09 | 1.76 | 1.04 | 1.19 | **0.111** |
| **Nested 1-GNN** (with DE) | **0.428** | **0.29** | **0.00265** | 0.00297 | **0.0038** | **20.5** | **0.00020** | **0.295** | **0.361** | **0.305** | **0.489** | 0.174 |
| **1-2-GNN** | 0.493 | **0.27** | 0.00331 | 0.00350 | 0.0047 | 21.5 | 0.00018 | **0.0357** | **0.107** | **0.070** | **0.140** | 0.0989 |
| **Nested 1-2-GNN** (no DE) | 0.454 | 0.308 | 0.00280 | 0.00278 | 0.0041 | 23.3 | 0.00029 | 0.349 | 0.281 | 0.395 | 0.307 | 0.0945 |
| **Nested 1-2-GNN** (with DE) | **0.437** | 0.278 | **0.00275** | **0.00271** | **0.0039** | **20.4** | **0.00017** | 0.252 | 0.265 | 0.241 | 0.272 | **0.0891** |
| **1-3-GNN** | 0.473 | 0.46 | 0.00328 | 0.00354 | 0.0046 | 25.8 | 0.00064 | 0.6855 | 0.686 | 0.794 | 0.587 | 0.158 |
| **Nested 1-3-GNN** (no DE) | 0.448 | 0.298 | 0.00276 | 0.00276 | 0.0040 | 22.0 | 0.00025 | 0.410 | 0.396 | 0.370 | 0.422 | 0.0936 |
| **Nested 1-3-GNN** (with DE) | **0.436** | **0.261** | **0.00265** | **0.00269** | **0.0039** | **20.2** | **0.00017** | **0.291** | **0.278** | **0.267** | **0.287** | **0.0879** |
| **1-2-3-GNN** | 0.476 | 0.27 | 0.00337 | 0.00351 | 0.0048 | 22.9 | 0.00019 | **0.0427** | **0.111** | **0.0419** | **0.0469** | 0.0944 |
| **Nested 1-2-3-GNN** (no DE) | 0.449 | 0.306 | 0.00282 | 0.00286 | 0.0041 | 22.0 | 0.00023 | 0.220 | 0.218 | 0.268 | 0.205 | 0.0975 |
| **Nested 1-2-3-GNN** (with DE) | **0.433** | **0.265** | **0.00279** | **0.00276** | **0.0039** | **20.1** | **0.00015** | 0.205 | 0.200 | 0.249 | 0.253 | **0.0811** |

In Table 6, we color the cell with light green if the NGNN (no DE) is better than the base GNN, and mark the cell with green if the NGNN (with DE) is additionally better than the NGNN (no DE). From the results, we can first observe that NGNNs (no DE) generally outperform the base GNNs, validating that even without any feature augmentation the NGNN framework still enhances the performance of base GNNs. Furthermore, we can observe that if NGNN improves over the base GNN, adding DE features could further enlarge the performance improvement by achieving the smallest MAEs among the three (i.e., base GNN, NGNN (no DE) and NGNN (with DE)). This demonstrates the usefulness of augmenting NGNN with DE features. Note that adding such DE features can be done simultaneously with the rooted subgraph extraction process, which only adds a negligible amount of time. Thus, augmenting NGNN with DE features is almost a free yet powerful operation to further enhance NGNN's power, which motivates us to make it a default choice of NGNN.