# OpenReview forum: "Nested Graph Neural Networks"
_NeurIPS.cc/2021/Conference — NeurIPS 2021 Poster_

### Official Review · Reviewer_etFC · 2021-07-14

**Rating:** 6
**Confidence:** 4

**Summary:**

This paper introduces Nested Graph Neural Networks (NGNNs), a novel kind of graph neural network that overcomes the limitations of typical message passing approaches by representing each node as a function of its nested subgraph, rather than its nested subtree.
The result is a model provably more expressive than 1-WL and 2-WL, at the cost of a constant increase in computational complexity (although this has practical limitations, as the authors point out).

NGNN works on two levels:

1. A "base GNN" is used to compute representations for each node, applying the GNN to the nested subgraph around the node and eventually pooling the subgraph to obtain the representation.

2. An "outer GNN" processes the resulting graph to obtain the final node representations, and a global pooling is optionally applied to get graph-level embeddings.

Therefore, NGNN can surpass 1-WL expressivity using only standard message-passing GNNs, making it very convenient and more computationally efficient than previous higher-order GNN variants.

The authors prove the expressive power of NGNN in distinguishing n-sized r-regular graphs.

The paper concludes with a thorough experimental analysis of NGNN on several benchmarks of graph classification and regression: QM9, ogb-molhiv, ogb-molpcba, and some TU datasets.
The method achieves state-of-the-art performance on ogb-molpcba and several QM9 subtasks.

**Limitations And Societal Impact:**

The limitations of the work are adequately addressed.
The paper is unlikely to have a negative societal impact.

**Main Review:**

**Originality**: the method is novel and smartly combines existing techniques, achieving impressive results. The work is contextualized well w.r.t. previous literature and highlights the similarities and advantages of NGNN.

**Quality**: the paper is technically sound. The claims are clearly stated at the beginning of the paper and methodically verified, with relevant proofs delegated to the Appendix. The goals of the experiments section are also clearly listed (which is something that one rarely sees in papers of this kind) and there is a final discussion that relates the results to the research questions.
The limitations of NGNN are discussed openly:

1. it is unclear whether NGNN is more powerful than 3-WL;
2. NGNN can only be used on graphs with up to 400 nodes on average, due to the memory limitations of GPUs.

**Clarity**: the paper is well written and organized. I have found only one minor typo: the comma after "because" on line 43 should be removed.
A few more details on how to extract subgraphs, in practice, would be needed to reproduce the results only from the paper, although this is only for the reader's convenience. All other experimental details are clearly described.

**Significance**: typical message-passing GNNs have significant limitations in terms of expressivity, and much of the research in the field is dedicated to designing more expressive GNNs. Most methods, however, resort to very expensive solutions (up to $O(n^3)$ complexity) which make them poorly applicable in real scenarios. NGNN overcomes the limitations of both message-passing GNNs and higher-order approaches. I can see how this paper could have a positive impact on the GNN community.
Also, the method proposed achieves the state of the art on a significant benchmark (ogb-molpcba) through a novel approach, which is an important result.
However, the other experimental results could be stronger and I ask the authors to comment on the points that I raise below, regarding the selection of baselines and the choice of datasets.

**Comments**: I have a few comments to strengthen the paper

- As a subjective observation, the description of the rationale for subgraph pooling in lines 69-76 could be more clear, as Figure 1 doesn't do a good job of showing the differences between how subtrees and subgraphs are constructed. One objection that at first comes to mind is that the base GNN is still working on subtrees, and the figure should show more clearly how this is, in fact, not the case (e.g., by coloring the nodes to make the differences more visually striking).
Since this is the core of the method, more effort should also be spent on its explanation in the text.

- How is subgraph extraction implemented in practice? The paper should describe more clearly why it is necessary to "materialize" the subgraphs in GPU memory, since it is the most evident shortcoming of NGNN. This would allow readers to understand the problem better and possibly come up with solutions.

- For OGB, how were the baselines selected?
The top-4 best results on ogb-molhiv are above 0.8 ROC-AUC, while all baselines here are below 0.8, making NGNN the best method due to this selection bias.

- The claimed improvements in performance on the TU datasets are often statistically insignificant (e.g., standard deviations up to 11.6 on MUTAG) and not very informative.
Since all graphs in OGB's graph-property prediction tasks are within the limit of 400 nodes on average, why didn't the authors consider ogb-ppa and ogb-code2 instead of TU?

- The code for experiments was not included in the supplementary material with the submission. It is important that the code is made available to reviewers prior to the final AC decision regarding this paper.

---
**After rebuttal**

My initial comments were minor and the authors have addressed most of them. Some issues remain about the computational complexity of the method, as also confirmed by the authors.

My rating remains unchanged.

**Time Spent Reviewing:**

6

---

> ### Author Response · Authors · 2021-08-10
> **Author Response**
>
> We thank the reviewer for their insightful review. In response to the reviewer’s questions and comments:
>
> - Re Q1: “Clarity: the paper is well written and organized. I have found only one minor typo: the comma after "because" on line 43 should be removed. A few more details on how to extract subgraphs, in practice, would be needed to reproduce the results only from the paper, although this is only for the reader's convenience.”
>
> We thank the reviewer for these comments. We will fix the typo, and give more details on how to implement the subgraph extraction in practice for a better reproducibility. We also worked hard to clean our code during the author response. It is now available at https://anonymous.4open.science/r/NestedGNN-BE66/README.md. We will release our code to the public too.
>
> - Re Q2: “As a subjective observation, the description of the rationale for subgraph pooling in lines 69-76 could be more clear, as Figure 1 doesn't do a good job of showing the differences between how subtrees and subgraphs are constructed. One objection that at first comes to mind is that the base GNN is still working on subtrees, and the figure should show more clearly how this is, in fact, not the case (e.g., by coloring the nodes to make the differences more visually striking).”
>
> Thanks for the suggestion. We acknowledge that Figure 1 is less illustrative than expected. We will make a better figure to illustrate the difference between rooted subtrees and rooted subgraphs and how the base GNN is really learning rooted subgraphs instead of rooted subtrees.
>
> - Re Q3: “How is subgraph extraction implemented in practice? The paper should describe more clearly why it is necessary to "materialize" the subgraphs in GPU memory, since it is the most evident shortcoming of NGNN.”
>
> For each root node, we extract (copy) its rooted subgraph from the original graph, and make it independent from other rooted subgraphs. After extracting all rooted subgraphs, the original graph becomes a large disconnected graph, where each component is a rooted subgraph for one original node. On this new graph, we perform the message passing and subgraph/graph pooling, and train the NGNN end-to-end. The “create_subgraphs()” function in “utils.py” defines the NGNN data structure. The reason why "materialize" each subgraph is mainly for batch-processing all the subgraphs simultaneously. Otherwise, we need to extract the subgraph on the fly for each node during the computation, resulting in a much higher latency.
>
> - Re Q4: “For OGB, how were the baselines selected? The top-4 best results on ogb-molhiv are above 0.8 ROC-AUC, while all baselines here are below 0.8, making NGNN the best method due to this selection bias.”
>
> As stated in Sec 5.2, we use *GNNs* achieving top places on the OGB graph classification leaderboard as the baselines. At the time of submission, DGN is the best GNN for ogb-molhiv, while other better methods on the leaderboard are based on fingerprints or improved Weisfeiler-Lehman kernel, which have limited performance on datasets other than ogb-molhiv (this might indicate ogbg-molhiv is particularly in favor of these traditional graph classification algorithms over GNNs). Thus, we exclude these methods in our baseline choices, in order to better compare NGNN with pure GNN methods.
>
> - Re Q5: “The claimed improvements in performance on the TU datasets are often statistically insignificant (e.g., standard deviations up to 11.6 on MUTAG) and not very informative. Since all graphs in OGB's graph-property prediction tasks are within the limit of 400 nodes on average, why didn't the authors consider ogb-ppa and ogb-code2 instead of TU?”
>
> The experiments on the TU datasets follow a reviewer’s suggestion in the last submission. We are aware that TU results can have large variances due to small sizes. Nevertheless, NGNN consistently shows improvements over base GNNs in all TU datasets, which could alleviate this issue. Further, please see our response to reviewer 5zao’s Q5, where we add another set of TU experiments with hyperparameter searching. Those results again demonstrate NGNN’s steady improvement over base GNNs.
>
> Although ogb-ppa has an average number of nodes less than 400, it has significant denser graphs with an average #edges larger than 2200. Therefore, our implementation of NGNN is still not scalable for it right now. For ogb-code2, it contains only tree-structured graphs, thus we expect NGNN to bring little gain to a standard message passing GNN which is already good at modeling rooted subtrees. We will consider doing an experiment to verify it in the revision.
>
> - Re Q6: “The code for experiments was not included in the supplementary material with the submission. It is important that the code is made available to reviewers prior to the final AC decision regarding this paper.”
>
> We now make the code available at https://anonymous.4open.science/r/NestedGNN-BE66/README.md. We include all the commands to reproduce our experiments in the README file.

---

### Official Review · Reviewer_5zao · 2021-07-15

**Rating:** 4
**Confidence:** 5

**Summary:**

This paper presents a framework that applies GNN to subgraphs around each node independently to enhance node representations, followed by a global pooling layer to get graph-level embedding. The framework can be applied to any GNNs. The paper also demonstrates it's relative improvement over several real-world datasets, although the improvement seems marginal. From theoretical side, the paper proved formally that this framework can distinguish k-regular graphs where most traditional message passing based neural network cannot.

**Ethical Concerns:**

Not applicable.

**Limitations And Societal Impact:**

Not applicable.

**Main Review:**

1. Originality: the idea of making use of subgraph to encode a better node representation is not new. In fact there some very similar papers that haven't been mentioned, and the architecture is exactly the same to their models except that the model mentioned in this paper can use any GNNs. One paper I remember is G-meta[1]. Please refer to their papers in detail and discuss the difference in terms of model architecture. The ability of using any type of GNNs seems a natural extension of their models. All in all the novelty is not enough considering the idea has been explored before.

2. Quality: the architecture and the subgraph information is solid to me. Including more information from subgraph can outperform original 1-hop based message passing model. However the information provided in experiment section is not complete. The hyperparameter tuning is not very clear to me, whether the number shown in the paper follows a fair evaluation should be stated clearly and also by providing a link to their code to check.

3. Clarity: the paper in general is well-written. The only part that is not clear to me is the "outer-GNN" mentioned in the paper. From my understanding the outer part is just a global pooling instead of a GNN model. Please explain this part clearly and if you only use a global pooling layer, please don't use the name "outer-GNN", instead use "global pooling" stage directly.

4. Significance: The result is kind of questionable from my end.

First, the TUDataset result is very different from the result in GIN paper, where GIN model has a lot better performance than the one shown in the paper. Maybe the experiment configuration is different. However, the paper mentioned that  "we uniformly use 4 message passing layers with 32 hidden dimensions each for all models" and "For NGNN, we uniformly use height-3 rooted subgraphs with mean pooling". This part sounds to me an unfair evaluation between NGNN and GNN models. I would question whether expand the pool of hyperparameter search will increase performance of GNN and decrease the performance of NGNN, as NGNN suffers more from overfitting than GNN. I suggest the author clearly state the pool of hyperparameters and provide a link to their code to have a look.

Second, the performance on molhiv and molpcba is also questionable. Comparing GIN and Nested-GIN I do see a 1% performance improvement, however the performance of Nested-GIN is not high enough comparing with other more recent baselines. This question whether Nesting GNN is the correct way to improve expressiveness and real-world performance. The ensemble result is not attractive to me, as this is not a fair evaluation to other methods. In face the author should present other models' ensemble results. On the other perspective, the large variance of the result shows the unstable training of nested GNN. The experiment section needs a huge improvement and fair evaluation.


[1] Graph Meta Learning via Local Subgraphs.  https://arxiv.org/pdf/2006.07889.pdf

**Time Spent Reviewing:**

3

---

> ### Author Response · Authors · 2021-08-10
> **Author Response**
>
> We thank the reviewer for their insightful review. We did our best to address the concerns. Specifically, we have cleaned our code, and now an anonymous code submission is available at https://anonymous.4open.science/r/NestedGNN-BE66/README.md. We also added several supplementary experiments to address the concerns on fairness of evaluation and significance of the results. The detailed responses are as follows.
>
> - Re Q1: “the idea of making use of subgraph to encode a better node representation is not new. In fact there some very similar papers that haven't been mentioned… One paper I remember is G-meta[1]. Please refer to their papers in detail and discuss the difference in terms of model architecture…. All in all the novelty is not enough considering the idea has been explored before.”
>
> The idea of using local enclosing subgraphs to represent *nodes and links* is indeed not new. For example, we have discussed such works including SEAL, k-hop GNN and MixHop in the related work. The mentioned paper G-meta[1] is also such a work. We will include G-meta in the revised version, too. However, the idea of using local subgraphs to enhance *whole-graph* representation learning is an original contribution of NGNN. NGNN for the first time incorporates the subgraph-based node representation learning into an end-to-end message-passing-based graph representation learning framework, which is a nontrivial extension to the existing subgraph-based works. Moreover, we have theoretically demonstrated the additional expressive power (going beyond 1-WL) brought by such a design. All these are not explored in previous works.
>
> - Re Q2: “The hyperparameter tuning is not very clear to me, whether the number shown in the paper follows a fair evaluation should be stated clearly and also by providing a link to their code to check.”
>
> Our experiments all follow a fair evaluation protocol (see our response to Q5 for details). We now provide our code for check, in which the README file contains all the commands for reproducing our experiments.
>
> - Re Q3: “ the paper in general is well-written. The only part that is not clear to me is the "outer-GNN" mentioned in the paper. From my understanding the outer part is just a global pooling instead of a GNN model. Please explain this part clearly and if you only use a global pooling layer, please don't use the name "outer-GNN", instead use "global pooling" stage directly.”
>
> We originally designed NGNN to be an extremely flexible framework---after the inner-GNN gets the final node representations, we allow the outer-GNN to further perform another few rounds of message passing between these node representations before the final pooling. However, we found that the additional message passing is not very useful in the used datasets. Thus, we only keep the pooling part, yet still call it an outer-GNN to keep the flexibility of our framework and also to reflect our core idea of nested GNN. We will make this clear in the revision.
>
>
> - Re Q4: “The result is kind of questionable from my end. First, the TUDataset result is very different from the result in GIN paper, where GIN model has a lot better performance than the one shown in the paper. Maybe the experiment configuration is different.”
>
> The GIN paper indeed uses a different experiment configuration from ours. Specifically, they use the validation set to select the early-stopping epoch and directly report the validation accuracy of 10-fold CV. This will overestimate the true performance on the test set, resulting in higher-than-normal accuracy numbers. In comparison, we strictly follow the train/val/test split, and report the test accuracy at the best validation epoch.
>
> - Re Q5: “the paper mentioned that "we uniformly use 4 message passing layers with 32 hidden dimensions each for all models" and "For NGNN, we uniformly use height-3 rooted subgraphs with mean pooling". This part sounds to me an unfair evaluation between NGNN and GNN models. I would question whether expand the pool of hyperparameter search will increase performance of GNN and decrease the performance of NGNN”
>
> Our comparison between NGNNs and GNNs follows a fair setting. Firstly, we use 4 message passing layers with 32 hidden dimensions for both GNNs and the base GNNs of NGNNs, so that NGNNs and GNNs have roughly the same number of parameters. Secondly, we uniformly use a subgraph height=3 for all NGNNs without tuning it because GNNs do not have such a hyperparameter. Besides, according to our discussion on subgraph height $h$ and base GNN layers $l$ in Appendix B, we would like an $l \geq h$ to more sufficiently absorb the whole-subgraph information, which is why we choose $h=l-1=3$ here.
>
> Nevertheless, we agree that using cross-validation to search these hyperparameters would be a better choice. Therefore, we conduct the following experiment. For GNNs, we search the number of message passing layers $l$ in {2,3,4,5}. For NGNNs, we similarly search the subgraph height $h$ in {2,3,4,5}, so that both NGNNs and GNNs can have equal-depth local receptive fields. For NGNNs, we always let $l=h+1$ instead of searching $l$ for each $h$, because that will make NGNNs have more hyperparameters to tune. In this *fair* setting, we record the test accuracy of each fold with the hyperparameter chosen based on the validation performance of this fold. Finally, we report the average test accuracy across 10 folds. The entire pipeline is based on PyTorch Geometric's benchmarking framework. The results are shown below (we compare each pair of GNN and its nested version, with the bold numbers indicating the better results between the two).
>
> |                  | D&D           | MUTAG         | PROTEINS      | PTC_MR        | ENZYMES       |
> |------------------|---------------|---------------|---------------|---------------|---------------|
> | GCN              | 0.716 ± 0.028 | 0.734 ± 0.108 | 0.717 ± 0.047 | 0.564 ± 0.071 | 0.273 ± 0.055 |
> | Nested GCN       | **0.763 ± 0.038** | **0.829 ± 0.111** | **0.733 ± 0.040** | **0.573 ± 0.077** | **0.312 ± 0.067** |
> | GraphSAGE        | 0.716 ± 0.030 | 0.740 ± 0.088 | 0.712 ± 0.052 | **0.570 ± 0.055** | **0.307 ± 0.063** |
> | Nested GraphSAGE | **0.774 ± 0.042** | **0.839 ± 0.107** | **0.742 ± 0.037** | **0.570 ± 0.059** | **0.307 ± 0.063** |
> | GIN              | 0.705 ± 0.039 | 0.845 ± 0.089 | 0.706 ± 0.043 | 0.512 ± 0.092 | **0.383 ± 0.064** |
> | Nested GIN       | **0.778 ± 0.039** | **0.879 ± 0.082** | **0.739 ± 0.051** | **0.541 ± 0.077** | 0.290 ± 0.080 |
> | GAT              | 0.710 ± 0.044 | 0.739 ± 0.107 | 0.720 ± 0.033 | **0.570 ± 0.073** | **0.302 ± 0.042** |
> | Nested GAT       | **0.760 ± 0.044** | **0.819 ± 0.102** | **0.737 ± 0.048** | 0.567 ± 0.081 | 0.295 ± 0.057 |
>
> As we can see, NGNNs still outperform their corresponding base GNNs in most cases, demonstrating that NGNNs can bring steady improvement to their base GNNs. The hyperparameter search will not affect NGNNs' advantages over GNNs.
>
> - Re Q6: “the performance on molhiv and molpcba is also questionable. Comparing GIN and Nested-GIN I do see a 1% performance improvement, however the performance of Nested-GIN is not high enough comparing with other more recent baselines….The ensemble result is not attractive to me, as this is not a fair evaluation to other methods. In face the author should present other models' ensemble results.”
>
> Firstly, we believe achieving SoTA results is not the only standard for evaluating a paper’s contribution. Very often, SoTA results on the leaderboard are driven by tricks, extensive hyperparameter search, data augmentation or pretraining, which can be unfair or misleading. For example, the current top 3 submissions on the ogbg-molpcba leaderboard (https://ogb.stanford.edu/docs/leader_graphprop/) are Graphormer, GINE+bot, and GINE + w/ APPNP, with 31.40, 29.94 and 29.79 test AP, respectively. However, Graphormer first pretrained its model on PCQM4M-LSC (the current biggest graph prediction dataset), GINE+bot (bag of tricks) uses a combination of seven tricks such as adversarial training to boost the performance, and GINE + w/ APPNP uses APPNP to enhance GINE+. In this sense, it is hard to evaluate a method’s true value by looking at the leaderboard only.
>
> Secondly, Nested GIN only uses a relatively weak base GNN here, yet already achieves competitive performance with SoTA methods. It is reasonable to expect NGNN combined with a stronger base GNN can achieve even better performance. Like the other tricks used by the leaderboard methods, we present the ensembled NGNN results only to show that NGNN itself can also be boosted to achieve SoTA results (our ensembled result on ogbg-molpcba of 30.07 ranks 1st at the time of submission).
>
> In summary, we believe all our experiments have firmly shown that NGNN brings consistent gains to its base GNNs in a plug-and-play way. In addition, with appropriate base GNNs and tricks, NGNN is able to compete with SoTA methods on the open leaderboard. Theoretically, NGNN also discriminates almost all $r$-regular graphs, making it more powerful than 1-WL and all message passing GNNs. Therefore, we would appreciate it if the reviewer could re-evaluate the significance of our work.

---

> ### Author Response · Authors · 2021-08-25
> **Would you mind letting us know if our response has addressed your concerns?**
>
> Dear reviewer 5zao,
>
> First of all, we would like to thank you again for the valuable feedback! As there is only about 1 week left for the discussion phase, we would greatly appreciate it if you could confirm whether our response has addressed your concerns.
>
> Please allow us to briefly summarize our previous response. In our initial response, we have
>
> 1. Clarified our difference to existing works using subgraphs to represent nodes/links, and pointed out the significance of using local subgraphs to learn whole-graph representations introduced by NGNN.
>
> 2. Submitted our code via an anonymous link for all reviewers to check (the commands to reproduce our experimental results are included in README.md).
>
> 3. Explained why our TU results are different from the GIN paper, and added a whole new set of experiments by searching hyperparameters for both GNNs and NGNNs on 5 TU datasets.
>
> 4. Explained why we believe merely evaluating a model by looking at the OGB leaderboard can be problematic.
>
> If you think there is still any other issue, please kindly let us know and we are more than happy to follow up with you before the end of the discussion phase. We believe NGNN makes a solid contribution to the field of studying more powerful GNNs than 1-WL. It is a plug-and-play framework that can consistently improve a base weak GNN's performance, despite that it may not lead to SoTA OGB results currently.
>
> Thanks,
>
> Authors

---

### Official Review · Reviewer_k3Ci · 2021-07-16

**Rating:** 7
**Confidence:** 4

**Summary:**

Since GNNs based on rooted subtrees are known with limited expressive power, this paper proposes to represent nodes with their rooted subgraphs. Its expressive power with respect to graph isomorphism testing is proven beyond 2-WL. Such a method is compatible to plug into popular GNNs to enhance their representation power. Enhanced GNNs show strong performances on graph classification and regression tasks.

**Limitations And Societal Impact:**

Yes.

**Main Review:**

This paper proposes an interesting method to define message passing on rooted subgraph, different from rooted subtrees. It is compatible with popular base GNNs. The method proves to go beyond 2-WL w.r.t. graph isomorphism testing. Real-world graph classification and regression tasks are conducted. It also presents necessary ablation study along with a synthetic experiment. This paper is well written.

Pros:

1. different from rooted subtrees and higher-order structures, this paper designs a more expressive gnn with easy-to-follow rooted subgraphs.
2. the proposed method is compatible to plug into popular but weaker GNNs and results in a guaranteed boost of expressive power. Extra computation time it incurs is reasonable.
3. distance encoding is natural in the NGNN framework as feature augmentation bringing excellent results.
4. weaker GNNs enjoy impressive performance boosts in real-world experiments.
5. the synthetic experiment in Appendix C confirms the theoretical result.

Concerns:

1. it would be better to try NGNNs on the ZINC dataset, where many popular papers proposing novel models present their results these days. It will bring more impact for this paper.
2. where is the sentence ''our initial analysis...'' in line 220 pointing to?
3. ablation study of distance encoding on graph classification tasks is missing. It is not sure whether the gain is from distance encoding or rooted subgraphs.

I suggest an accepting score.

**Time Spent Reviewing:**

4

---

> ### Author Response · Authors · 2021-08-10
> **Author Response**
>
> We thank the reviewer for their positive review. Our response is as follows.
>
> - Re Q1: “it would be better to try NGNNs on the ZINC dataset, where many popular papers proposing novel models present their results these days. It will bring more impact for this paper.”
>
> Thanks for the suggestion. Due to the time constraint of the author response, we are not able to finish implementing a pipeline for ZINC experiments. We will add ZINC experiments in the revision.
>
> - Re Q2: “where is the sentence ''our initial analysis...'' in line 220 pointing to?”
>
> We did some preliminary analysis comparing the expressive power of 3-WL and NGNN, yet did not get a conclusion. Thus we did not include the analysis in the paper. We will make this clear.
>
> - Re Q3: “ablation study of distance encoding on graph classification tasks is missing. It is not sure whether the gain is from distance encoding or rooted subgraphs.”
>
> The ablation study on distance encoding is included in Appendix E. The experiments show that NGNN alone without DE is also powerful, while combining NGNN with DE can further improve the performance.

---

### Official Review · Reviewer_6fjM · 2021-07-16

**Rating:** 4
**Confidence:** 5

**Summary:**

This paper proposes a new architecture to strengthen message passing GNNs. Since the weaknesses of GNNs usually come from  the unidentifability of nodes, the authors suggest to first preprocess given graphs and add each node a new initial attribute, and then apply the outer GNN. The new node attributes come from another message passing GNN (called based GNN) applied to a subgraph of a certain depth (or height according to the paper's terminology) around each node. Then they prove that the new architecture can distinguish regular graphs with probability $1-o(1)$ (over the choice of a random regular graph), and so the new model is more expressive than MPNNs, since MPNNs can't. They finally evaluate their model in various settings.

For me the contributions of the paper are: (1) proposing nested GNNs (NGNNs); (2) showing that NGNNs can distinguish almost all regular graphs; (3) and evaluating the new model through various experiments.

**Limitations And Societal Impact:**

Yes

**Main Review:**


Strengths:

(1) The paper is really easy to understand. I could understand the manuscript at the first read.

(2) They supported their theory with nice experiments.

(3) The new theoretical result is of potential interest to the graph learning community since it provides significant contribution to the problem of learning regular graphs.

Weaknesses/comments/questions:

(0) A number of important previous works are missing. For example, in the context of counting substructures, people also considered strengthening GNNs with subgraph pooling [1]. In [1] the authors consider a similar setting where MPNNs are strengthened with pooling on a neighborhood around each node (similar idea), and instead of having base GNN, they consider Relational Pooling (RP). Please provide explanations why/how your paper is different from theirs, and compare the results (also see [4,5]).
Also, if  we have random node IDs, then GNNs can approximate any functions (e.g., see [2]). So you need to also mention those papers in your paper and explain what is your contribution. The bound you get for the complexity is also a special case of what is appeared in a preprint [3]. The referred  papers below are only examples and you can find more related papers by tracking these examples (and references therein). However, I haven't seen your theorem on distinguishing regular graphs in any previous papers, so the paper contributes on the problem of learning with regular graphs. I suggest to rewrite some parts of the paper and mention that you essentially have an original theoretical result on learning with regular graphs, and also explain/discuss/mention all the related works.  I believe the paper has good contributions, but it is still vague how it is related to the previous works, it has missing references (just I provided a few examples below), and extensive revision is required.

 [1] Chen, Zhengdao, et al. "Can graph neural networks count substructures?." arXiv preprint arXiv:2002.04025 (2020).

 [2] Abboud, Ralph, et al. "The surprising power of graph neural networks with random node initialization." arXiv preprint arXiv:2010.01179 (2020).

[3] Tahmasebi, Behrooz, and Stefanie Jegelka. "Counting Substructures with Higher-Order Graph Neural Networks: Possibility and Impossibility Results." arXiv preprint arXiv:2012.03174 (2020).

[4] Lou, Zhaoyu, et al. "Neural Subgraph Matching." arXiv preprint arXiv:2007.03092 (2020).

[5] Ying, Rex, et al. "Hierarchical graph representation learning with differentiable pooling." arXiv preprint arXiv:1806.08804 (2018).

(1) The lack of detailed motivation behind the theorem: since your theorem just concerns regular graphs, there should be a reasonable motivation for why we need to  do classification/regression on regular graphs (please provide examples that regular graphs arise, like in molecules, etc.).

(2) I think if the given graphs are strongly regular, then the model fails to distinguish them. Is it true? Please provide more discussion about this fact right after the main theorem because it could help the reader and it's also related to the future works.


(3) The proposed method is a feature augmentation trick, however, I couldn't find any literature review on that topic in the related works section. Please add related papers and also discuss how your model is different from theirs.



(4) While the main body of the paper is quite understandable, the appendix is not. Specially, the proof of your main theorem is not clear at all. There is a reference  to a prior work which is not clear to me unless I read the reference. Can you add a few sentences about steps 1-3 in appendix (line 637) to make the proof more readable?  There is no page restriction for the appendix.



(5) In Equation 3, are the functions $U_t,M_t$ shared among nodes? Or each node has its specific function? I guess they share, but it is not so clear in the paper.


(6) Figure 2 is great for understanding your model. Can put it earlier in the paper and give more references to it within the text?

(5) Line (229): I guess $O(cn^3)$ --> $O(cn^2)$.

(6) In the main theorem, can we have $\epsilon \to 0$, so we only need one layer message passing? How does the distinguishing error in Theorem 1 (i.e., the term $1-o(1)$) depend on $\epsilon,n,r,...$? I believe one important question is the term $o(1)$ in your proof. Can you quantify that in your theorem?    Please quantify the dependence of the $o(1)$ bound in your result to $\epsilon,n,r$ to allow comparison and help readers.  Also, please provide a few sentences as proof sketch right after the main result.


(7) I cannot understand why the new algorithm increase the computational load just by a constant? Do you mean $c$ in line 229 is $O(1)$? In your setting with regular graphs, we could even have $c = r^h = O(n^{0.5})$ if we apply those $r,h$ in Theorem 1.  Is it true? Hence, your algorithm does not always increase the computational load by a constant factor. It can grow with $n$. Please clearly mention in in the paper.

 --------------------------------

After rebuttal:

The authors mostly agreed on the comments, and explained some of them, but to me some of the questions are still unanswered, or it is still not clear what exactly do they want to add to the paper after revision. I decided to keep my rating.


**Time Spent Reviewing:**

10

---

> ### Author Response · Authors · 2021-08-10
> **Author Response 1/2**
>
> We thank the reviewer for their thoughtful, comprehensive and constructive review as well as for acknowledging our theoretical contribution. We will add all the missing references to our paper, and discuss/compare them with our work.
>
> - Re Q1: “A number of important previous works are missing. For example, in the context of counting substructures, people also considered strengthening GNNs with subgraph pooling [1]… Please provide explanations why/how your paper is different from theirs, and compare the results (also see [4,5]).”
>
> Thanks for introducing these relevant works. We will discuss/compare them in our revised paper. In [1], relational pooling to encode the subgraph around each node is adopted. Computing relational pooling means that one needs to impose each possible order of the nodes in the subgraph, encode the subgraph under each possible order and then pool the encoded representations into a single subgraph representation. This way is much more computationally complicated than our simple GNN model. We do not clearly see how [4] and [5] are relevant as [4] is to perform subgraph matching instead of entire graph representation learning. [5] can perform entire graph representation learning. However, the hierarchical pooling proposed by [5] is not to encode the subgraph around each node but to group nodes into several clusters and perform pooling for each cluster.
>
> Nevertheless, we would like to empirically compare our method with [1] and [5] that also target entire graph representation learning. We compare [1] using the common datasets used between [1] and our paper, i.e., ogbg-molhiv and QM9. The results on *ogbg-molhiv* are as follows:
>
> |                   | Validation | Test       |
> |-------------------|------------|------------|
> | Deep LRP-1-3      | 81.31±0.88 | 76.87±1.80 |
> | Deep LRP-1-3 (ES) | 82.09±1.16 | 77.19±1.40 |
> | Nested GIN        | 83.17±1.99 | **78.34±1.86** |
> | Nested GIN (ens)  | 80.80±2.78 | **79.86±1.05** |
>
> \
> The results on *QM9* are as follows.
>
> |                             | Deep LRP-1-3 | Deep LRP-5-1 | Nested 1-GNN | Nested 1-2-GNN | Nested 1-3-GNN | Nested 1-2-3-GNN |
> |-----------------------------|--------------|--------------|--------------|----------------|----------------|------------------|
> | $\mu$                       | 0.399        | **0.364**    | 0.428        | 0.437          | 0.436          | 0.433            |
> | $\alpha$                    | 0.337        | 0.298        | 0.29         | 0.278          | **0.261**      | 0.265            |
> | $\varepsilon_{\text{HOMO}}$ | 0.00287      | **0.00254**  | 0.00265      | 0.00275        | 0.00265        | 0.00279          |
> | $\varepsilon_{\text{LUMO}}$ | 0.00309      | 0.00277      | 0.00297      | 0.00271        | **0.00269**    | 0.00276          |
> | $\Delta \varepsilon$        | 0.00396      | **0.00353**  | 0.0038       | 0.0039         | 0.0039         | 0.0039           |
> | $\langle R^2 \rangle$       | 20.4         | **19.3**     | 20.5         | 20.4           | 20.2           | 20.1             |
> | ZPVE                        | 0.00067      | 0.00055      | 0.00020      | 0.00017        | 0.00017        | **0.00015**      |
> | $U_0$                       | 0.590        | 0.413        | 0.295        | 0.252          | 0.291          | **0.205**        |
> | $U$                         | 0.588        | 0.413        | 0.361        | 0.265          | 0.278          | **0.200**        |
> | $H$                         | 0.587        | 0.413        | 0.305        | **0.241**      | 0.267          | 0.249            |
> | $G$                         | 0.591        | 0.413        | 0.489        | 0.272          | 0.287          | **0.253**        |
> | $C_v$                       | 0.149        | 0.129        | 0.174        | 0.0891         | 0.0879         | **0.0811**       |
>
> As we can see, NGNN compares favorably with Deep LRP [1] on both ogbg-molhiv and QM9. We also add an experiment comparing Nested GraphSAGE with DiffPool [5] (with GraphSAGE as its convolution block) on TU datasets. We search the subgraph height $h$ in {2,3,4,5} for Nested GraphSAGE and search the number of layers in {2,3,4,5} for DiffPool. The results are as follows.
>
> |                  | D&D               | MUTAG             | PROTEINS          | PTC_MR            | ENZYMES           |
> |------------------|-------------------|-------------------|-------------------|-------------------|-------------------|
> | DiffPool         | **0.778 ± 0.039** | 0.830 ± 0.055     | **0.742 ± 0.045** | 0.529 ± 0.055     | **0.338 ± 0.062** |
> | Nested GraphSAGE | 0.774 ± 0.042     | **0.839 ± 0.107** | **0.742 ± 0.037** | **0.570 ± 0.059** | 0.307 ± 0.063     |
>
> We find our Nested GraphSAGE has competitive performance to DiffPool. Despite that the two methods do not have a clear winner, NGNN has the advantage that the operations are performed on the sparse graph structure, while DiffPool need to transform the adjacency matrix into a dense format, which might cause problems for large graphs.
>
>
>
> - Re Q2: “if we have random node IDs, then GNNs can approximate any functions (e.g., see [2]). So you need to also mention those papers in your paper and explain what is your contribution. ”
>
> Using random node features indeed can improve the GNN representation power. However, one single generation of random node features makes GNN lose the nice permutation invariance property, which causes severe generalization issues. Therefore, random node features should be generated multiple times for each training example, which induces rather slow model training convergence. This is indeed observed in the experiments of [2]. Our method obviously adopts a fundamentally different way to improve GNN power, by using a two-level nested GNN framework. Therefore, our model does not have such training convergence issues. We will discuss the relation between our work and [2] in the revised version of the paper.
>
> - Re Q3: “The lack of detailed motivation behind the theorem: since your theorem just concerns regular graphs, there should be a reasonable motivation for why we need to do classification/regression on regular graphs (please provide examples that regular graphs arise, like in molecules, etc.).”
>
> We focus our discussion on regular graphs because it is an important category of graphs where standard GNNs cannot represent well. Using 1-WL or standard message passing GNNs, any two n-sized r-regular graphs will have the same representation. Thus, if we can prove a GNN discriminates almost all r-regular graphs, it is guaranteed to surpass 1-WL in expressive power. Moreover, the extra power over regular graphs is just what we need to characterize in order to distinguish a more powerful GNN from a standard message passing GNN. The reason is because, all graphs (even an irregular one) will become attributed regular graphs after performing node color updates in 1-WL till convergence (Arvind et al. 2019). The expressive power over regular graphs can be directly applied to those cases, i.e., the convergent status of 1-WL test. When message passing/1-WL converges, more powerful GNNs like ours can further distinguish these graphs, while standard GNNs may fail.
>
> “On Weisfeiler-Leman invariance: subgraph counts and related graph properties, ” Arvind et al. 2019
>
> - Re Q4: “I think if the given graphs are strongly regular, then the model fails to distinguish them. Is it true?”
>
> Reviewer 6fjM is very insightful. This is correct.
>
>
>
> - Re Q5: “The proposed method is a feature augmentation trick, however, I couldn't find any literature review on that topic in the related works section. Please add related papers and also discuss how your model is different from theirs.”
>
> We actually discuss in detail those approaches that leverage deterministic feature augmentation (Sec. 4 paragraphs 3 and 4), including [30, 31, 32, 41,42], and random feature augmentations (Sec. 4 paragraphs 2), including [40,41]. We have detailed their differences from our work. For the other relevant works introduced by the reviewer, we would like to include our discussion on them in the revised version.
>
>
>
> - Re Q6: “While the main body of the paper is quite understandable, the appendix is not. Specially, the proof of your main theorem is not clear at all... Can you add a few sentences about steps 1-3 in appendix (line 637) to make the proof more readable?”
>
> Yes. We can do that in the revised version. The key idea of Lemma 1 is to consider the configuration model of a regular graph. We analyze the subtree rooted a node in the regular graph. From the root node to the nodes of more height in the subtree, we check how the edges that come from the nodes of a certain height L are allocated among the nodes of height L+1. Doing careful statistical analysis we find that when the height L lies in a certain range (from about 0.5 (log n)/(log(r-1)) to 2/3(log n)/(log(r-1)), where r is the node degree), most of the edges will attach to different nodes of height L+1. This observation is formally summarized in Lemma 1.
>
>
> - Re Q7: “In Equation 3, are the functions  shared among nodes?”
>
> Yes. The subgraph encoding Eq.3 is shared among nodes.
>
> - Re Q8: “Figure 2 is great for understanding your model. Can put it earlier in the paper and give more references to it within the text?”
>
> Yes. We will put Figure 2 to an earlier position.
>
> - Re Q9: “Line (229): I guess $O(cn^3)$ → $O(cn^2)$.”
>
> The complexity of Nested 1-2-3-GNN (or Ring-GNN/PPGN) is indeed $O(n c^3)$, because we first extract subgraphs of size $c$ for all $n$ nodes, and then applies 1-2-3-GNN to all these subgraphs (each takes $O(c^3)$).

---

> > ### Author Response · Authors · 2021-08-10
> > **Author Response 2/2**
> >
> > - Re Q10: “In the main theorem, can we have $\epsilon \rightarrow 0$, so we only need one layer message passing? How does the distinguishing error in Theorem 1 (i.e., the term $1-o(1)$) depend on $\epsilon, n, r...$? … Please quantify the dependence of the  bound in your result to  to allow comparison and help readers. Also, please provide a few sentences as proof sketch right after the main result.”
> >
> > Yes. We just need one layer message passing, which is also demonstrated empirically in Appendix C, Figure 3 (right).
> >
> > The more subtle distinguishing error follows  $n^{-0.5 + \epsilon}$ + $n^{3/2-(\epsilon^2/3) \log n / \log\log n}$. In practice, we choose a fixed $\epsilon$ and enlarge n to infinity, and then there will be almost no error. An empirical demonstration of this result is given in Appendix C Figure 3. We did not provide a proof sketch because of the page limitation. In the final version, if more pages are allowed, we can definitely provide some proof sketch.
> >
> > - Re Q11: “I cannot understand why the new algorithm increase the computational load just by a constant?”
> >
> > The constant times higher complexity requires us to bound the size of the extracted rooted subgraphs. If the subgraph size needs to grow with $n$ (like what is suggested by the reviewer in the regular graph case), then indeed the computation increase is not by a constant. But in practice, we only use a small subgraph height $h=3$ or $h=4$, which effectively bounds our subgraph size and encourages NGNN to learn local patterns. We will make this more clear in the paper.

---

> ### Author Response · Authors · 2021-08-25
> **Would you mind letting us know if our response has addressed your concerns?**
>
> Dear reviewer 6fjM,
>
> First of all, we would like to thank you again for the valuable feedback! As there is only about 1 week left for the discussion phase, we would greatly appreciate it if you could confirm whether our response has addressed your concerns.
>
> Please allow us to briefly summarize our previous response. In our initial response, we have
>
> 1. Added discussion on the suggested related work such as relational pooling [1], neural subgraph matching [4] and random node initialization [2], and compared with the suggested baselines Deep LRP [1] and DiffPool [5] on three datasets, ogbg-molhiv, QM9 and TU.
>
> 2. Motivated our choice of analyzing GNN's expressive power for regular graphs (Theorem 1).
>
> 3. Added sentences to explain Theorem 1 and its proof better.
>
> 4. Clarified a few misunderstandings such as relation to feature augmentation techniques, weight sharing of base GNNs, time complexity of Nested 1-2-3-GNN, constant factor higher complexity, etc.
>
> We made a great effort in writing the author response. If you think there is still any other issue, please kindly let us know and we are more than happy to follow up with you before the end of the discussion phase.
>
> Thanks,
>
> Authors

---

### Official Review · Reviewer_MGt6 · 2021-07-17

**Rating:** 6
**Confidence:** 4

**Summary:**

This paper proposes Nested GNN, which encodes each node based on the subgraph rather than the subtree (as what GNN does) around it. The main difference between NGNN and GNN is that subgraph captures more information than the subtree, such as the interactions between a node's neighbors. The whole-graph representation can be pooled from each node representation, and can be applied in tasks such as graph classification. From theoretical perspective, this paper studies the limitations of expressive power of GNN and show that NGNN is more powerful. The proposed method shows good performance on multiple datasets.

**Limitations And Societal Impact:**

My concern about the this method is the computational cost. It applies base GNN for subgraph around each node, meanwhile the classical GNN learns node representation in a one-pass fashion. The authors mentions such limitation as in their Section 5.3, that NGNN does not scale to graph datasets with average node number > 400. I am wondering could you perform the base GNN step for all nodes in parallel, since extracting subgraph and train GNN on it should be independent for each node,

**Main Review:**

This paper studies the limitation of expressive power of GNN and proposes NGNN, which encodes each nodes representation based on the subgraph rather than the subtree around it. The point and the case study of when GNN would fail is valid, and the proposed method seems to perform well and is shown can discriminate r-regular graphs. I am a bit concerned about the novelty of the method since the first step (base GNN) seems to be just a way to learn each node representation (within the subgraph, while GNN learn it within the whole graph), and the second step is a standard graph pooling. I also have the following questions that hope the author could address:

1. Do you have any guidance on choosing the subgraph height at the base GNN step? I noticed the current experiment uniformly select height = 3, I think this is a parameter that should be related with the graph size or could be tuned in a principled way.

2. The main concern is about the computational cost (also mentioned in the limitation section). I noticed that the average size of graphs in the datasets are relatively small. Have you tried or implemented the base GNN step in parallel? Also, related the first question, if your graph size is small, then extracting a height-3 subgraph rooted in a node would not have too much difference with the whole graph. Would be interesting to see the performance on graph with larger size,

3. The base GNN layers allow a node (v) to have different representations within difference rooted subgraphs (call the root u), which I view it as a "contextual" representation of node v. Currently the method uses the contextual representation of v to pool the representation of the rooted node u. Besides that, could you make use of the contextual representation of v in different rooted subgraph in other ways, for example to encode v itself when it is the root?


**Time Spent Reviewing:**

5

---

> ### Author Response · Authors · 2021-08-10
> **Author Response**
>
> We thank the reviewer for their insightful review. In response to the reviewer’s questions and concerns:
>
> - Re Q1: “I am a bit concerned about the novelty of the method since the first step (base GNN) seems to be just a way to learn each node representation (within the subgraph, while GNN learn it within the whole graph), and the second step is a standard graph pooling.”
>
> The reviewer's description of NGNN is correct. Our NGNN framework is designed to be a simple but effective way to improve a base message passing GNN’s expressive power in a plug-and-play way. The novelty lies exactly in using a base GNN to learn node representation within its rooted subgraph. Such a design enables a node representation to encode its rooted subgraph more than merely a rooted subtree as in standard message passing GNNs, and has a good theoretical property for distinguishing regular graphs.
>
> - Re Q2: “Do you have any guidance on choosing the subgraph height at the base GNN step? I noticed the current experiment uniformly select height = 3, I think this is a parameter that should be related with the graph size or could be tuned in a principled way.”
>
> Yes, this parameter should be related to the dataset and graph size. Theoretically, to distinguish regular graphs, our Theorem 1 provides a guidance (i.e., choose h = 0.5log(n)/log(r-1)). As shown in Appendix Figure 3, for 100-node 3-regular graphs, an h=3 is enough, and for 1000-node graphs, an h=4 is enough. For practical datasets, we can tune it like how we tune the number of layers in a standard GNN. Nevertheless, we find h=3 to be generally a good choice.
>
> - Re Q3: “The main concern is about the computational cost (also mentioned in the limitation section). I noticed that the average size of graphs in the datasets are relatively small. Have you tried or implemented the base GNN step in parallel?”
>
> Yes. We implemented the base GNN in a parallel way. Specifically, all the rooted subgraphs of a graph are loaded to the GPU memory to be processed in parallel. Please refer to our anonymous code submission at https://anonymous.4open.science/r/NestedGNN-BE66/README.md for more details.
>
>
> - Re Q4: “Also, related the first question, if your graph size is small, then extracting a height-3 subgraph rooted in a node would not have too much difference with the whole graph. Would be interesting to see the performance on graph with larger size,”
>
> Since all the rooted subgraphs are loaded to the GPU and processed together, the GPU memory restricts the applicability of our current implementation of NGNN to larger graphs (over 400 nodes). In fact, if we reduce the batch size, e.g., to let a batch only contain one graph, then copying all the rooted subgraphs of this graph to GPU is still feasible even this graph has more than 400 nodes. But we find this results in significantly lower performance than a normal mini-batch training. In the future, to resolve this issue, we plan to implement the base GNN in a sequential or distributive way, so that different graphs of a batch can be loaded to different GPUs.
>
> - Re Q5: “The base GNN layers allow a node (v) to have different representations within difference rooted subgraphs (call the root u), which I view it as a "contextual" representation of node v. Currently the method uses the contextual representation of v to pool the representation of the rooted node u. Besides that, could you make use of the contextual representation of v in different rooted subgraph in other ways, for example to encode v itself when it is the root?”
>
> Thanks for the interesting idea. We very much agree that the contextual representations can be better used. We will try to better leverage both the contextual and root representation of a node in our future work.

---

### Author Response · Authors · 2021-08-10
**Code available now.**

Dear reviewers,

We thank all your effort and constructive feedback in reviewing our paper. In response to many reviewers' request of code, we now make it available at https://anonymous.4open.science/r/NestedGNN-BE66/README.md. We also provide an alternative link to directly download the zip file at https://www.filemail.com/d/fjymreqjsxnampw. We hope this code submission can alleviate some concerns about implementation details, reproducibility and evaluation fairness.

Thanks,

Authors

---

### Decision · Program_Chairs · 2021-09-27

**Decision:**

Accept (Poster)

**Comment:**

This work proposes a way to overcome some of the known expressivity issues of permutation invariant message-passing GNN. It does so by learning node features by applying a GNN on a subgraph around each node.

The reviewers agreed that the paper is well written and that it makes an interesting theoretical contribution. Moreover, the empirical evaluation (taking also into account the results presented during the rebuttal period) demonstrates that the proposed architecture can bring practical benefits in terms of accuracy. The main catch seems to be that the computational/memory complexity of the proposed method might be prohibitive for large graphs.

The reviewers were divided in their evaluations. The main issue expressed was that the proposed idea is not surprising given the literature and that a thorough discussion of relevant works was missing from the submitted paper. Even though these concerns are justified, it seems that the specific idea proposed has not been considered/tested before. Moreover, the authors acknowledged and explained the connections to previous works in the rebuttal period. Thus, I don't see an issue with accepting the work under the condition that the camera-ready version is updated appropriately.